# Linking Crystallographic Orientation and Ice Stream Dynamics: Evidence from the EastGRIP ice core

Nicolas Stoll[1,2,3], Ilka Weikusat[1,4], Daniela Jansen[1], Paul Bons[4], Kyra Darányi[1,4,5], Julien Westhoff[6], María-Gema Llorens[7], David Wallis[8], Jan Eichler[1,9], Tomotaka Saruya[10], Tomoyuki Homma[11], Sune Rasmussen[6], Giulia Sinnl[6], Anders Svensson[6], Martyn Drury[5], Frank Wilhelms[1,12], Sepp Kipfstuhl[1], Dorthe Dahl-Jensen[6,13], and Johanna Kerch[1,12]

[1]Department of Geosciences, Alfred Wegener Institute Helmholtz Centre for Polar and Marine Research, Bremerhaven, Germany
[2]Department of Environmental Sciences, Informatics and Statistics, Ca'Foscari University of Venice, Venice, Italy
[3]Department of Earth and Space Sciences, University of Washington, Seattle, USA
[4]Geoscience Department, Eberhard Karls University, Tübingen, Germany
[5]Department of Earth Sciences, Utrecht University, Utrecht, The Netherlands
[6]Physics of Ice, Climate and Earth, Niels Bohr Institute, University of Copenhagen, Copenhagen, Denmark
[7]GEO3BCN-CSIC, Lluís Solé Sabarís s/n, 08028 Barcelona, Spain
[8]Department of Earth Sciences, University of Cambridge, Cambridge, CB2 3EQ, UK
[9]Laboratoire de Géologie de Lyon: Terre, Planètes, Environnement (LGL-TPE), ENS Lyon, Université Claude Bernard Lyon 1, CNRS, Villeurbanne, France
[10]National Institute of Polar Research, Tokyo 190-8518, Japan
[11]Nagaoka University of Technology, 1603-1 Kamitomioka-machi, Nagaoka 940-2188, Japan
[12]Geoscience Centre, University of Göttingen, Göttingen, Germany
[13]Centre for Earth Observation Science, University of Manitoba, Winnipeg, Canada

**Correspondence:** Ilka Weikusat (ilka.weikusat@awi.de)

**Abstract.** A better understanding of glacial ice flow and how it is influenced by internal deformation is required to improve the projections of future sea-level rise in a warming climate. Especially large ice streams, the main contributors to solid ice discharge to the ocean, still require more observational data to be represented sufficiently in numerical ice-sheet models. The East Greenland Ice-core Project (EastGRIP) successfully drilled the first continuous deep ice core through an active ice stream, the Northeast Greenland Ice Stream (NEGIS), focusing on investigating the dynamical processes that lead to its exceptionally high velocity. Here, we show Crystallographic Preferred Orientations (CPO) data in 5–15 m depth resolution throughout the 2663 m core, down to bedrock, to determine the deformation regimes in this ice stream, complemented by grain-size and borehole temperature profiles for context. A broad single-maximum CPO pattern is present in the upper 200 m caused by overlying snow and ice layers. Below, a crossed girdle CPO is observed for the first time in a deep ice core and we discuss possible formation mechanisms. Between 500 and 1230 m of depth, we observe a vertical girdle CPO indicative of along-flow extensional deformation. A complementary simple-shear component could explain the CPO between 1230 and 2500 m, a vertical girdle with horizontal maxima of varying strength. Close to bedrock, large, amoeboid-shaped grains and a multi-maxima CPO indicate migration recrystallisation due to high temperatures close to the pressure melting point. Complementary conductivity data further suggest an undisturbed stratigraphy until at least 104 ka b2k and microstructural data suggest even

15  older ice from the Eemian. A comparison with other deep ice cores from Greenland and Antarctica shows the uniquely fast development of CPO at shallow depths in the EastGRIP ice core due to its location in an area of high strain rates while the grain-size profile with depth remains similar to less dynamic sites confirming that it is mainly governed by the varying purity of ice deposited during varying climatic conditions. We further show that the overall plug flow of NEGIS is characterised by many small-scale variations, which remain to be considered in ice-flow models.

## 1  Introduction

Melting and solid ice discharge are the two main processes resulting in the mass loss from Greenland and Antarctica. Ice streams, which are characterised by a river-like structure of localised high flow velocities, contribute most to solid ice discharge and thus play a crucial role regarding future sea level rise (e.g., Bamber et al., 2000; van den Broeke et al., 2009; Margold et al., 2015; IPCC, 2022). However, ice-stream dynamics, including the internal deformation, are insufficiently understood (IPCC, 2013; Nick et al., 2013; Stokes et al., 2016). Therefore, the physics of ice streams must be investigated further to enable better projections of global sea level rise. Large-scale ice-flow models usually ignore the mechanical anisotropy of ice (Winkelmann et al., 2011) or assume (scalar) enhancement factors to approximate the effects of anisotropy in the bulk behaviour (e.g., Russell-Head and Budd, 1979; Rathmann et al., 2022). As it is only possible to measure ice flow velocity on the surface, except for very few locations with long-term accessible boreholes, there are no direct observations to constrain ice behaviour deeper in the ice column in an ice stream. Thus, the relative contributions to fast ice flow from internal deformation and basal sliding can only be estimated by inverse modelling, relying on standard descriptions of the rheological behaviour of ice. It is further unclear how and where the horizontal movement over the bed translates into shearing over bed.

Internal deformation is a significant component of ice flow, and its relative importance depends on the basal conditions. However, direct observational data from within the ice volume, i.e. from a deep ice core, are scarce due to the high costs and effort involved. To study the internal deformation in an ice stream, physical properties, such as the microstructure and Crystallographic Preferred Orientations (CPO) of ice crystals (also called crystal-orientation fabric or, in short, fabric), are examined and improve our understanding of the rheological behaviour of ice. A better description of ice rheological behaviour can then be implemented in large-scale flow models to advance our understanding of ice dynamics. Modelling ice CPOs and respective deformation regimes has advanced over the recent decades, becoming a powerful tool for investigating the dynamic behaviour of ice at different scales (e.g., Montagnat et al., 2014b; Richards et al., 2023). However, objectives and implemented processes usually differ from small-scale to large-scale flow models (e.g., Azuma and Higashi, 1985; Alley, 1992; Piazolo et al., 2019; Lilien et al., 2021; Llorens et al., 2022; Rathmann and Lilien, 2022; Richards et al., 2023; Ranganathan and Minchew, 2024). Information on the anisotropy in ice streams could be deduced to a certain extent by geophysical remote sensing methods, such as seismic (e.g., Smith et al., 2017; Pearce et al., 2024) and radar (Jordan et al., 2022; Zeising et al., 2023; Gerber et al., 2023; Nymand et al., 2024) measurements. The most detailed ground-truth data are required from ice cores to verify the results of these remote sensing data in addition to improving modelling approaches.

Accurate information on the ice anisotropy and microstructural properties, such as grain size, can be derived via polarised light microscopy measurements of the c-axes orientations on consecutive thin-section samples from deep ice-core drillings. Measurements have been established utilising automated fabric analysers, typically at depth intervals of 10 to 150 m (e.g., Wang et al., 2003; Montagnat et al., 2014a; Fitzpatrick et al., 2014; Weikusat et al., 2017; Voigt, 2017). So far, most deep ice cores have been drilled at locations with low ice flow velocities, such as ice domes or divides, to guarantee an undisturbed record of climate signals (e.g., Petit et al., 1999; Watanabe et al., 2003; EPICA Community Members, 2004). Thus, direct observations of ice-sheet CPOs and microstructure were focused on these regions. However, the need for data from more dynamic regions is evident. The investigation of microstructural processes influenced by crystallographic preferred orientation, such as strain localisation and shear layers (e.g., Bons and Jessell, 1999; Carreras, 2001; Adam et al., 2005; Llorens et al., 2016a; de Riese et al., 2019), largely unexplored in ice, would further benefit from data at high spatial resolution on anisotropy and deformation in an ice stream.

To investigate ice-stream dynamics by direct observation from within the ice, the East Greenland Ice-core Project (EastGRIP, main drilling between 2016 and 2023) retrieved the first continuous deep ice core through the central part of an active ice stream, the Northeast Greenland Ice Stream (NEGIS) (Fig. 1). NEGIS is Greenland's most significant ice stream draining 12% of the ice sheet (Rignot and Mouginot, 2012). Jansen et al. (2024) showed that NEGIS was dynamic in the Holocene and has only been established in its current form 2000 years ago. Further, along-flow extension and compression is more difficult in the ice stream centre than the shear margins (Gerber et al., 2023).

Initial microstructural data from EastGRIP have assisted in investigating the localisation of impurities in the ice microstructure (Stoll et al., 2021a, 2022; Bohleber et al., 2023; Stoll et al., 2023) further studying the complicated interplay between impurities and the ice microstructure in regards to deformation and ice viscosity (e.g., Jones and Glen, 1969; Paterson, 1991; Eichler et al., 2017; Stoll et al., 2021b), in reconstructing the original orientation of the drilled core (Westhoff et al., 2021), and in understanding the birefringent radar echo patterns caused by CPO (Gerber et al., 2023).

We present the first-ever study of CPO patterns with depth throughout a site in the central part of a several-kilometre-wide ice stream derived via thin-section measurements from a deep ice core supplemented by grain size and electrical-conductivity data. Complete measurements of 0.55 m pieces, i.e. entire "bags", of the ice core every 5–15 m exceed the semi-continuous sampling established in prior deep ice-core studies. The closely spaced samples enable us to quantify gradual changes in CPO with depth, providing a solid observational platform to build robust deformation models showing the changes in dominant deformation kinematics, conditions and mechanisms as a function of depth inside NEGIS.

Here, we focus on the investigation of the large-scale (hundreds of m) and mesoscale (m) changes of CPO with depth and derive the second-order orientation tensor's eigenvalues to explore the CPO and ice anisotropy with depth. We further compare the CPO development to other ice cores from Greenland and Antarctica aiming to highlight the unique characteristics of ice streams by considering different dynamical settings.

## 2   Methods

### 2.1   The EastGRIP ice core

The EastGRIP deep drilling succeeded in retrieving the first continuous ice core through an active ice stream. The drilling site is located at 75°38.16' N and 35°59.35' W (Fig. 1a), 2708 m a.s.l (May 2024), roughly 10 km away from the shear margins of NEGIS (Fig. 1b and c), the most significant ice stream in Greenland in terms of ice transport towards the margin (Fahnestock et al., 1993). The surface ice flow velocity at EastGRIP is ~55 m/yr (Hvidberg et al., 2020), but the velocity, location and flow dynamics of NEGIS probably changed throughout the Holocene (Franke et al., 2022; Jansen et al., 2024). Upstream of EastGRIP, the surface velocity and width of NEGIS are smaller while it widens into faster flow downstream of the drill site (Fig. 1a) (Hvidberg et al., 2020). The site is characterised by an ice thickness of approximately 2667 m and an annual mean surface temperature of -28.7°C (Vandecrux et al., 2023).

Electromechanical drilling with the established Danish Hans-Tausen-style drill retrieving 98 mm diameter cores started in 2016. Drilling had to be stopped in 2020 and 2021 due to the COVID-19 pandemic. Drilling resumed in 2022, and bedrock, consisting of water-saturated fine-grained sediment, was reached in August 2023 at a depth of 2667.7 m. The bubble lock-in depth at EastGRIP is between 58 and 61 m of depth (Westhoff et al., 2023). Ice temperature (Fig. 1) is approximately -32°C to a depth of 1200 m. Temperature increases below this with gradually increasing thermal gradients to 2000 m where the temperature is -25°C. There is a constant thermal gradient to the final measurement at 2665 m, just above the base of the ice, where the temperature is -2.47°C, which is slightly below the pressure melting point.

### 2.2   Sample preparation

At the EastGRIP drill site, roughly every 5–15 m of depth, we cut 0.55 m long ice core pieces parallel to the core axis into six sections of 92 x 70 mm ("vertical sections"). In-situ core orientation, i.e. the azimuth angle, is not recorded directly during drilling as oriented drilling is still challenging. While this lack of orientation is not a constraint on obtaining results from the data analysis, it may limit the interpretation. The core azimuth with reference to geographic coordinates could only be reconstructed at specific depths using visual stratigraphy (Westhoff et al., 2021).

Unless indicated otherwise, the presented observations were carried out on vertical sections. This enables the continuous analysis of the variations in CPO with depth within 55 consecutive cm. Roughly every 100 m "horizontal sections" were taken from core volume samples of 100 mm length, providing snapshots of the microstructure orthogonal to the core axis, thus enhancing the representation of three-dimensional shape of crystals with two-dimensional measurements.

For a combined analysis of microstructure and CPO, samples were cut into thick and thin sections with thicknesses of 130–160 mm and 0.3 mm, respectively. Remaining pieces were used for measurements of impurity distribution in the microstructure (e.g., Stoll et al., 2021a, 2022, 2023; Bohleber et al., 2023). The samples were glued onto clean glass plates with water droplets and microtomed with a Leica microtome sledge. A sharp blade enables micrometre-precise polishing via a micrometre screw, adjusting the distance between the blade and the stage. Each sample was left to sublimate under controlled conditions at the ambient temperature in the trench (-28 to -18°C) for the required time, usually ca. 1 hour, to increase the visibility of grain

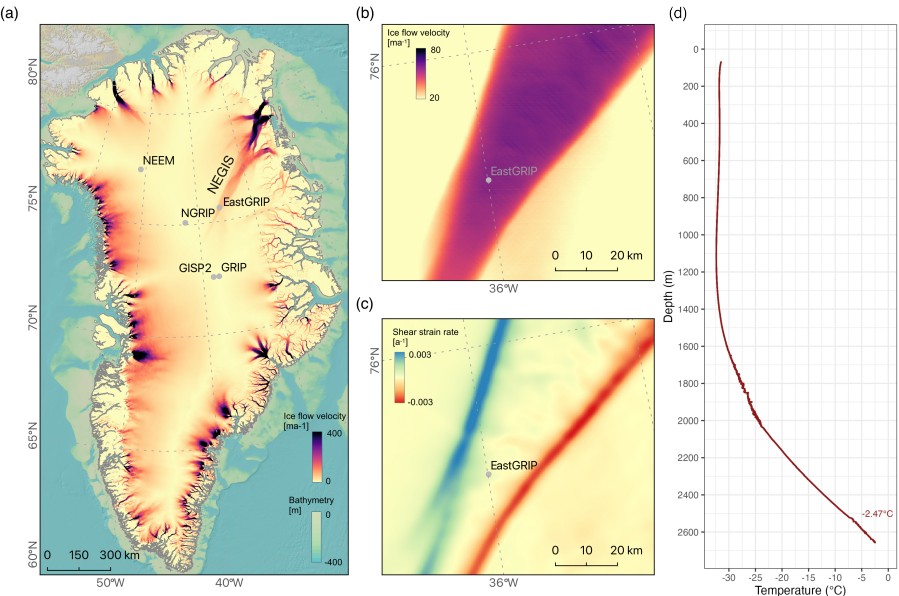

**Figure 1.** a) Overview of the surface ice flow velocity of Greenland (MEaSUREs ice velocity data set (Joughin et al., 2010a, b)). NEGIS and some deep ice coring sites are indicated. b) Detailed view of NEGIS surface ice flow velocity close to EastGRIP and the thereof c) derived shear strain rates in the horizontal plane. d) Temperature profile of the borehole derived from borehole logging on May 17, 2024. Data from July 9, 2023, is shown below 2610 m because the logger could not go deeper in 2024 due to a glycol cavity at this depth. The deepest measurement is -2.47°C, slightly below the pressure melting point.

boundaries as they develop into sublimation grooves (e.g., Weikusat et al., 2009) and smooth the surface roughness affected by the polishing.

### 2.3 CPO and grain size measurements with the fabric analyser

We used the automatic fabric analyser G50 manufactured by Russel-Head Instruments in the field shortly after drilling. The instrument automatically measures the CPO (Wilson et al., 2003) inside each thin section. We analysed 1153 vertical and 66 horizontal sections over five field seasons (2017–2019, 2022–2023) from 165 continuous 55 cm ice-core sections. Due to varying ice-core quality, volume samples, and time constraints, the remaining measurements come from less continuous sections. The measurements cover an ice depth range within the ice stream from 111 m to 2663 m, which is almost the depth of the bedrock. Ice from the brittle zone (approximately 650–950 m (Westhoff et al., 2022)) was measured a year later to avoid samples breaking easily before relaxation. Between 2120 and 2417 m, no CPO data were measured due to the time delay inflicted by the COVID-19 pandemic.

    The fabric analyser uses polarised light microscopy to derive the orientation of the c-axis by using the birefringence of polarised light in optically anisotropic media. Thus, it is possible to measure one part of the full crystal orientation, i.e. the

orientation of the main crystallographic axis, the c-axis. One standard measurement with a 20 $\mu$m spot size of a sample with dimensions of 92 x 70 mm takes roughly 45 minutes.

## 2.4   Fabric-analyser data processing

Raw image data were manually corrected to exclude artificial ice crystals that originate from the preparation process of the thin sections (Fig. A1). The corrected data are analysed with the software *cAxes* (Eichler, 2013). Threshold criteria regarding,

for example, minimum crystal sizes (500 px equaling 0.2 $mm^2$), misorientation (1°), and sample quality, were applied. The derived data contain information about physical properties, such as c-axis distribution and grain size. Grain area is calculated by transforming the pixels of single grains from the previously derived grain-boundary network into $mm^2$ yielding more exact results than older techniques, such as measuring the longest grain diameter or only a certain percentage of the largest grains. We here display the most common parameters to enable an overview of the core regarding deformation regimes.

## 2.5   Eigenvalues and fabric patterns

To analyse the shape and strength of the measured fabric data, the eigenvalues of the second-order orientation tensor for each sample are calculated. The measured c-axis orientations in polar coordinates are converted into Cartesian coordinates followed by the determination of the orientation tensor via standard structural geology methods (Wallbrecher, 1986). The c-axes distribution can be displayed as an ellipsoid with the eigenvectors representing the three orthogonal unit vectors along its

axes originating from the centre of the ellipsoid (Woodcock, 1977). The lengths of the axes are represented by the invariant eigenvalues ($e_1, e_2, e_3$), which are usually normalised ($\lambda_1$, $\lambda_2$, and $\lambda_3$). They obey the conditions $\lambda_1+\lambda_2+\lambda_3=1$ and $\lambda_1 \leq \lambda_2 \leq \lambda_3$. The dominant fabric type can be determined using the eigenvalues according to the following rules:

  – random fabric: $\lambda_1 \approx \lambda_2 \approx \lambda_3$, shape of a sphere;

  – single-maximum: $0 \leq \lambda_1 \approx \lambda_2 \leq \frac{1}{6}$ and $\frac{2}{3} \leq \lambda_3 \leq 1$, shape of a prolate ellipsoid;

– girdle fabric: $\lambda_1 < \lambda_2 \approx \lambda_3$, shape of an oblate ellipsoid.

However, the meaningfulness of second-order orientation tensor eigenvalues is limited due to their inability to differentiate between certain fabric types. For example, discriminating between multi-maxima and isotropic distributions is not possible. Thus, utilising additional data representations, such as (contoured) stereographic projections (provided in the supplement), are required to fully evaluate CPO patterns.

## 2.6   Dielectric profiling

Dielectric profiling (DEP) enables, before cutting and processing the core, the fast and non-destructive scanning of both, the electric conductivity and permittivity. At EastGRIP, DEP was performed on-site shortly after drilling with the device introduced by Wilhelms et al. (1998) to quickly locate positions of volcanic events and changes in chemical constituents. Conductivity in ice is mainly impacted by the acidity and the salt and ammonia concentrations (Moore et al., 1992). A detailed description of

the instrument, the established procedure at EastGRIP, and DEP data can be found in Mojtabavi et al. (2020). We here extend
       the DEP record to the deepest 260 m.

## 2.7    Electrical conductivity measurements

We made electrical conductivity measurements (ECM) with the technique originally described by Hammer (1980) with details
given for the current setup by Mojtabavi et al. (2020). The ECM signal is related to the acidity of the core, and the high-
resolution signal is beneficial for identifying volcanic signals (peaks in ECM) and layers with high $NH_4$ concentration (troughs
      in ECM). The records are used for peak and pattern matching, where sequences of peaks are assumed to represent the same
      sequence of events in the matched cores. The signal's relation to the absolute acidity is largely uncalibrated. Still, as the
      same method and setup have been used for NorthGRIP, NEEM and EastGRIP, the records are comparable in this regard. Data
      processing was carried out as described in Mojtabavi et al. (2020).

## 2.8    Chronology

Annual-layer identification is possible in at least the upper parts of the EastGRIP ice core but has so far only been completed for
the most recent 3.8 ka b2k (b2k: thousands of years before 2000 CE) (Sinnl et al., 2022). For most of the core, the Greenland Ice
Core Chronology 2005 (GICC05; Vinther et al. (2006), Rasmussen et al. (2006), Andersen et al. (2006), Svensson et al. (2006))
which is based on annual-layer identification in the DYE-3, GRIP, and NorthGRIP ice cores, has instead been applied to the
EastGRIP ice core based on the assumption that peaks and common patterns in the ECM and DEP data represent isochrones.
      The tools and methodology are described in Seierstad et al. (2014), and here, we extend the work of Mojtabavi et al. (2020)
      and Gerber et al. (2021), which provided similar transferred time scales for EastGRIP reaching back to 15 kyr to 49.9 ka b2k,
      respectively. Here, we only describe where our procedure or results differ from what has been described in these papers.
         Two observers independently aligned the ECM and DEP records of EastGRIP and NorthGRIP. Only minor differences
related to ambiguous peak shapes etc. were found between the two sets of match points, which were then combined to one
      set of match points. The result is 108 tie points between EastGRIP and NorthGRIP for the period older than 49.9 ka b2k
      (corresponding to EastGRIP depths of 2117 m and below). More details can be found in the appendix (A2).
         The ECM and DEP records can be matched with high confidence to and including Greenland Interstadial (GI) 23.1 at about
      104 ka b2k (Fig. 2). The short-lived GI-23.2 is well-resolved in the NorthGRIP ice core but is missing in both NEEM and
EastGRIP. A less certain but still credible match of GI-24 allows us to extend the time scale to about 108 ka b2k, but below
      this, the resemblance of the signals between cores drops, and several possible alignments of the records are possible. We thus
      refrain from assigning ages to the part of the record below GI-24.
         In the deepest section (below 2550 m), we observe a lack of sharp features and note that the transitions into and out of
      interstadials become more gradual, often spanning 0.5-1 m, adding to the alignment uncertainty of the oldest sections. The
annual layer thickness is about 11 cm just below the firn, 2 cm at 50 ka b2k, and just a few mm at 100 ka b2k, leading to
      a correspondingly large variation in time scale-transfer uncertainty when measured in years. The interpolation uncertainty is
      mainly controlled by the distance to the nearest match point, and the largest uncertainty is therefore expected in the middle of

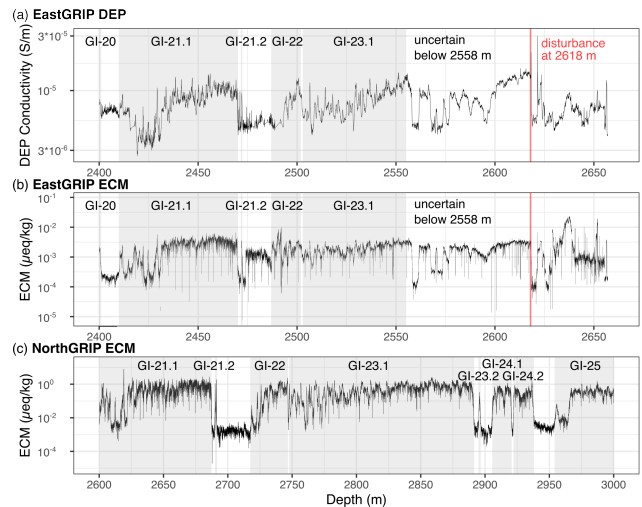

**Figure 2.** a) DEP and b) ECM data from the deepest 260 m compared to c) NorthGRIP ECM (Rasmussen et al., 2013) data. Greenland Interstadials (GI) are indicated after Rasmussen et al. (2014). The EastGRIP time scale is uncertain below 2558 m. S/m is siemens per metre.

the longest stadials (e.g., GS-18 and GS-19.1) and within the middle of GI-23.1, where we did not identify any match points. It is thus not possible to derive a simple estimate of the time scale-transfer uncertainty, and we urge users to be cautious when interpreting offsets between features found in the NorthGRIP, NEEM, and EastGRIP records.

## 3 Results

### 3.1 CPO patterns in the EastGRIP ice core

The c-axis distributions, displayed as a selection of representative pole figures in Figure 3 (all pole figures are provided in the supplements), reveal a pattern that has not been observed so far in a deep ice core. To enable the interpretation of the present CPO patterns, we must classify them and their respective depth regimes (Table 1). Most patterns are subject to gradual transitions and that the chosen depth regimes are inevitably subjective, approximations, and biased by the available data and can not be regarded as clearly defined.

The first 85 m, starting at an absolute depth of 111 m, show a weak broad single-maximum CPO. With depth, this CPO pattern strengthens and c-axes are orientated more towards the vertical which could be interpreted as weak girdles.

Between 196 and 294 m, we describe, for the first time in natural or experimentally deformed ice, crossed girdle CPOs (Fig. 4). Crossed girdle CPOs are defined by a meeting of the 'limbs' of circle girdles forming type I (Christie, 1963; Hara et al., 1973), type II (Sylvester and Christie, 1968; Lister, 1977), and transitional stage. Type I is characterised by two girdles meeting at some distance from each other while being connected by a single girdle. Type II is characterised by the direct meeting of the two girdles thus 'crossing' at one point (Schmid and Casey, 1986). We observe both types as well as transitional stages

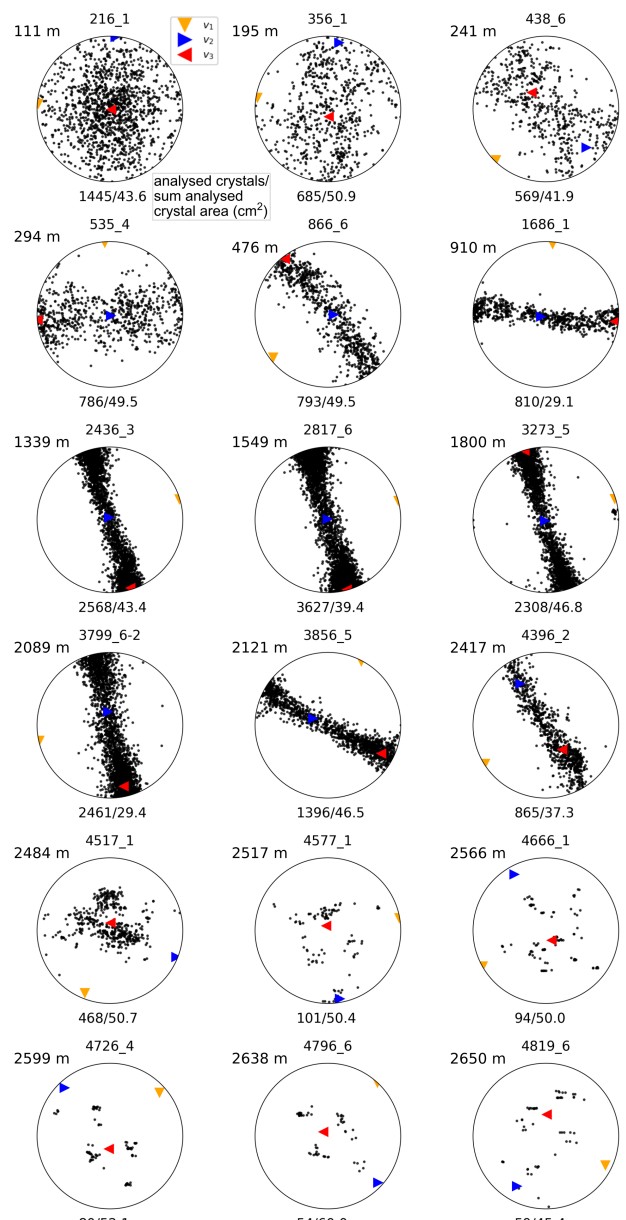

**Figure 3.** Representative selection of CPO patterns at EastGRIP displayed as equal-area lower hemisphere projections. Depth, bag number and section, eigenvectors, number of analysed crystals, and the sum of the analysed crystal area ($cm^2$) are indicated. The original azimuthal orientation is not preserved. All pole figures are provided in the supplements.

(Fig. 4 and supplements) and indicate them with skeletal outlines in Figure 4. Crossed girdles are strongest between 205 and 242 m. They vary in type and strength from sample to sample. The observed 'limbs' are differently pronounced and, for type II, vary in their meeting point in regards to the vertical axis. Despite a high sampling resolution of measurements up to every

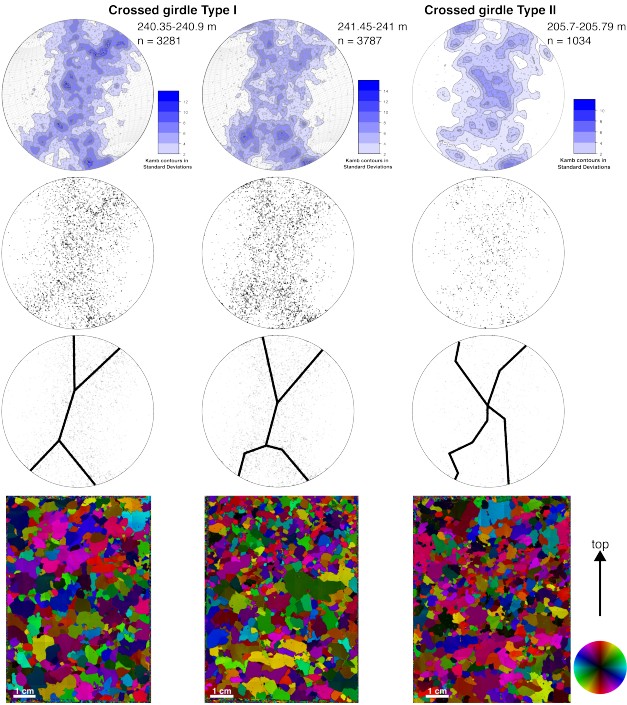

**Figure 4.** Representative examples of crossed girdle CPOs in the EastGRIP ice core and respective microstructure (c-axis orientations are indicated by the colour wheel; top points towards the surface of the ice sheet). Displayed are in down-wards direction equal-area lower hemisphere Kamb contour plots (Kamb, 1959), point plots, and fabric skeletons constructed from contoured c-axes plots. The sample from 205.7 m depth represents a crossed girdle type II. The two examples of crossed girdle type I are compiled of six consecutive thin sections over 55 cm to increase visibility. Note the differences in the legends.

metre in this depth regime, no continuous CPO development trend is visible. Between 294 and 500 m, the crossed girdle CPO (pronounced at e.g., 366 and 398 m; see supplements) transitions into a broad, vertical girdle CPO.

The vertical girdle CPO is fully formed from roughly 500 m of depth. It increases in strength, i.e. the girdle gets narrower, down to 1230 m. The vertical girdle often shows horizontal maxima of varying strength, which are fully established below 1394 m and occur down to 2418 m. Around 2418 m, grain sizes are large and the number of measured c-axes thus decreases resulting in less pronounced CPOs.

Between 2500 and 2663 m, the c-axes exhibit a multi-maxima CPO (Fig. 3 and 5). Most samples exhibit four to five maxima,
which are centred around the vertical axis. Centimetre large crystals with amoeboid shapes and curved grain boundaries are abundant at this depth. However, the number of measured crystals (21–168) is usually still sufficient to identify CPO patterns.

At the depth interval of 2608–2618 m, crystals are several centimetres in length (see section 3.4), and therefore do not have a sufficient number of crystals to measure in the samples. We combined data from six adjacent samples to yield suitable statistics with several hundreds of data points.

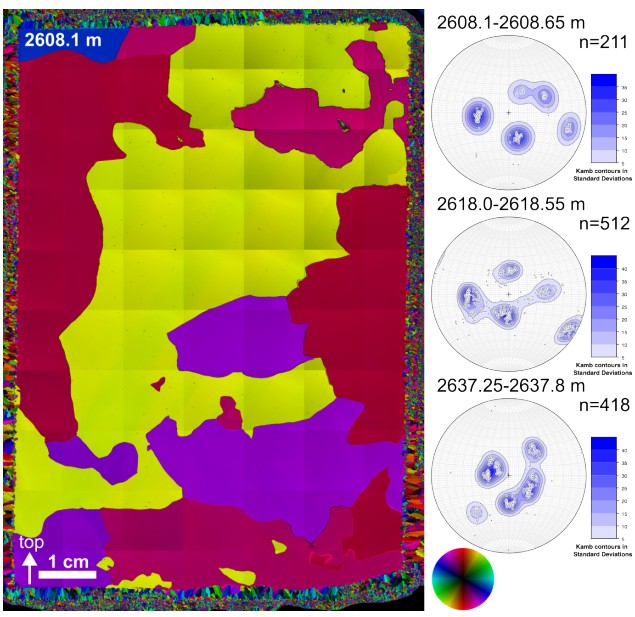

**Figure 5.** Example of the microstructure at 2608.1 m in c-axis orientation colour scale (see colour wheel on bottom right). Adjacent are three representative multi-maxima CPOs with Kamb contours (Kamb, 1959) compiled of six consecutive thin sections over 55 cm. The middle plot displays the interface between Eemian and glacial ice, probably due to a stratigraphic disturbance. Projections and annotation as in Figure 3 excluding the rotation; note the slight differences in the legends.

**Table 1.** Approximate depth regimes of the present CPO patterns at EastGRIP. In reality, CPO patterns are subject to gradual transitions and the specified depths are chosen for simplicity. The mentioned transition zones are not displayed separately, but visible in the supplement material .

| Approximate depth (m) | Age range b2k (ka) | CPO pattern |
|---|---|---|
| 111–196 | 0.7–1.5 | Broad Single-Maximum |
| 196–294 | 1.5–2.3 | Crossed Girdle |
| 294–493 | 2.3–4.1 | Crossed Girdle-Vertical Girdle Transition |
| 493–1230 | 4.1–11.5 | Vertical Girdle |
| 1230–1394 | 11.5–15.3 | Vertical Girdle with developing Horizontal Maxima |
| 1394–2500 | 15.3– 89.6 | Vertical Girdle with Horizontal Maxima |
| 2500–2663 | older than 89.6 , possibly ~120 | Multi-maxima |

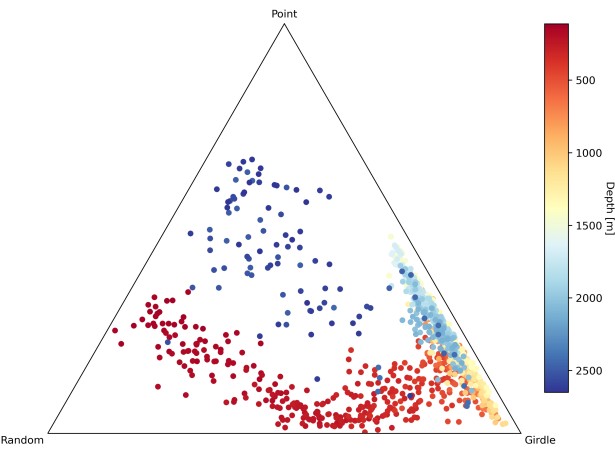

**Figure 6.** Triangular (ternary) diagram displaying the CPO data classified between random (R), point (P) and girdle (G) CPO following Vollmer (1989, 1990). The indices P, G and R are calculated from the grain-weighted eigenvalues (P = $\lambda_3$-$\lambda_2$, G = 2*($\lambda_2$-$\lambda_1$), R = 1 - P - G). Every marker represents one vertical thin section; depth is indicated by colour.

## 3.2 Patterns of eigenvalues with depth

We show eigenvalues as mean values per 92 x 70 mm thin section between 111 and 2663 m depth. Figure 6 displays it as a ternary diagram displaying the ratios of the end-members random, point, and girdle CPO summing to a constant (Vollmer, 1989, 1990). These indexes are analogous to calculations following Woodcock (1977) and display the differences between eigenvalues and not their ratios in a closed way with three, instead of two, end-members. For EastGRIP, this shows the change in CPO with depth transforming from random with a point component to a strong girdle. Close to bedrock, it develops towards somewhere between point and random representing multi-maxima CPOs.

The classic representation of the three eigenvalues with depth is shown in Figure 7a. Between 111 and 500 m, $\lambda_1$ decreases constantly and stays close to 0 until a depth of 2400 m. $\lambda_2$ and $\lambda_3$ start around 0.25 and 0.5, respectively, at 111 m (broad vertical maximum CPO), and meet at 0.4 at 250 m correlating with the fully formed crossed girdle CPO. They separate until 550 m (vertical girdle CPO); from whereon a wavy eigenvalue pattern of $\lambda_2$ and $\lambda_3$ occurs (Fig. 7a). The amplitude of this wavy pattern is usually between 0.1–0.2 and the wavelengths range from dozens to hundreds of meters of depth. $\lambda_2$ and $\lambda_3$ are similar at 650, 720, 930, 1110, 1370, and 1895 m. At 2400 m, $\lambda_1$ starts to increase up to 0.2 and $\lambda_2$ decreases to minimum values of 0.15. Below 2550 m, $\lambda_1$ and $\lambda_2$ decrease and $\lambda_3$ increases towards 0.8 correlating with the multi-maxima CPO. Eigenvalue variability is much greater at this depth than in shallower ice.

The Woodcock parameter (Fig. 7b) lies in the interval 0–1 and 1–∞ for girdle and unimodal CPOs, respectively. In EastGRIP, the Woodcock parameter fluctuates between 1 and 6.2 in the shallowest analysed 70 m and decreases with depth. Between 180 and 2480 m, it is below 0.5. We detect slightly greater values between 228 and 250 m of depth. Below 2400 m, the Woodcock

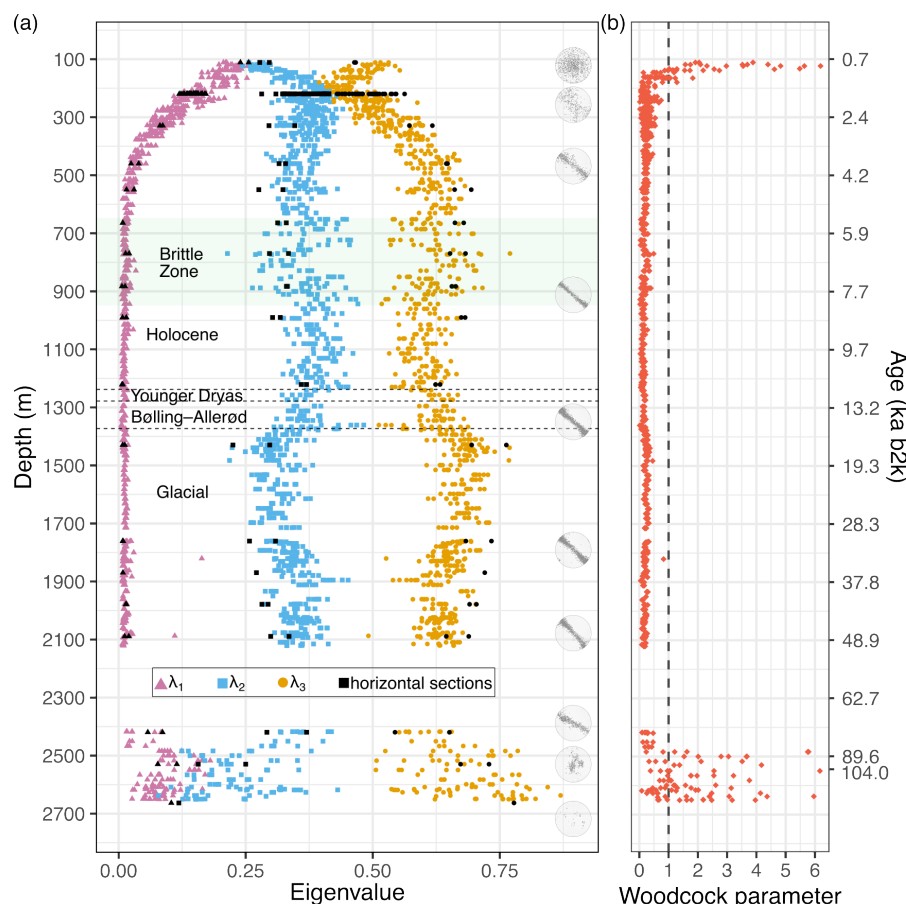

**Figure 7.** a) Orientation tensor eigenvalues of thin sections (9.2 x 7 cm) as derived by the fabric analyser and representative CPOs. Coloured eigenvalues are from vertical sections orientated parallel to the core axis; black symbols represent horizontal sections perpendicular to the core axis. Brittle zone (Westhoff et al., 2022) and climatic periods are indicated. b) Woodcock parameter, values above 1 represent unimodal CPOs; values below 1 indicate girdle CPOs. One outlier of 20.5 at 135.5 m of depth was excluded for better visibility. Shown values are grain area-weighted. Novel ECM and DEP data extend the time scale with certainty until 104 ka b2k.

parameter increases and is often greater than 1. It sometimes exceeds 5 below 2483 m, correlating with the multi-maxima CPO. We excluded one outlier of 20.5 at 135.5 m in Figure 7b, which is probably a measurement artefact.

## 3.3 Microstructure overview: grain size and shape

The mean grain area, from hereon referred to as grain size, is displayed in Figure 8. Figure 5, 9, and 10 provide an overview of grain shape and grain size distribution. A companion study will provide an in-depth, quantitative analysis of high-resolution microstructure data utilising Large Area Scanning Macroscope (LASM) data (Krischke et al., 2015).

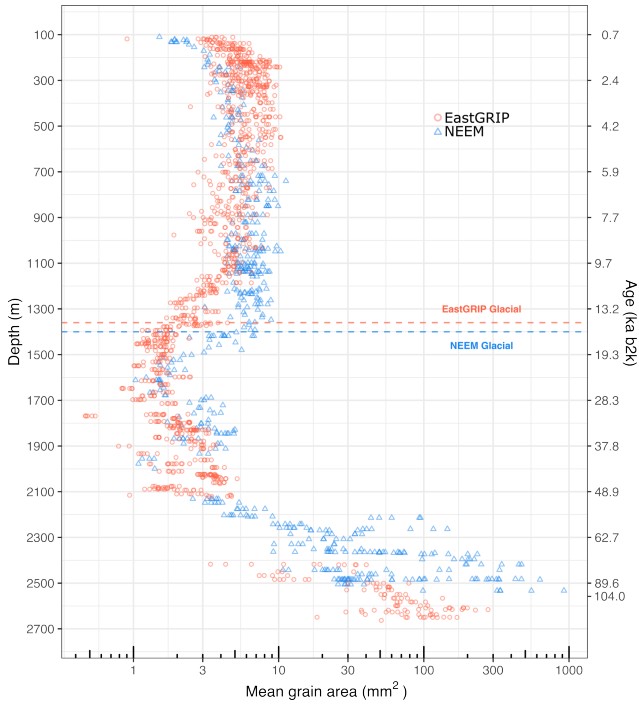

**Figure 8.** Comparison of the mean grain size between EastGRIP and NEEM (Montagnat et al., 2014a) derived by automated fabric analysers. Ice below the dotted lines is from the Last Glacial and the largest grains likely indicate Eemian ice.

In the upper 550 m, grain size reaches up to 10 $mm^2$, and grains are usually characterised by lobate grain boundaries and
equidimensional to oblate shapes (Fig. 9a). Grains tend to become more horizontally elongated with depth, and the grain size decreases to 3–6 $mm^2$ at 1000 m, increases slightly until 1100 m, and decreases to about 2–3 $mm^2$ at 1300 m. The grain size is between 0.5 and 5 $mm^2$, mainly fluctuating around 1–2 $mm^2$ with an increasing trend with depth throughout the glacial until 2500 m. Further, grain size layering is common, especially in fine-grained bands, and grain shapes are usually rectangular with straight grain boundaries. In the deepest ~150 m grain size strongly increases up to 280 $mm^2$ (Fig. 8) and amoeboid grain
shapes with bulging grain boundaries dominate (Fig. 5 and 9a). Different types of subgrain boundaries were observed at all depths. Grain size distributions are skewed in all samples with a peak at finer grain sizes and a long tail extending to coarser sizes (Fig. 9b). Mean values are greater than the medians. In the deepest samples, the distribution is strongly skewed but the small number of large grains hamper statistical analyses.

The range of the mean grain size measured per thin section varies strongly throughout the core. The range is relatively
large in the Holocene (Fig. 8), spanning up to 10 $mm^2$ between the smallest and largest mean values at similar depths. In the glacial, the variability is smaller, i.e. between 0.5 and 5 $mm^2$. At 2410 and 2618 m, there is a large variability between adjacent samples (Fig. 10).

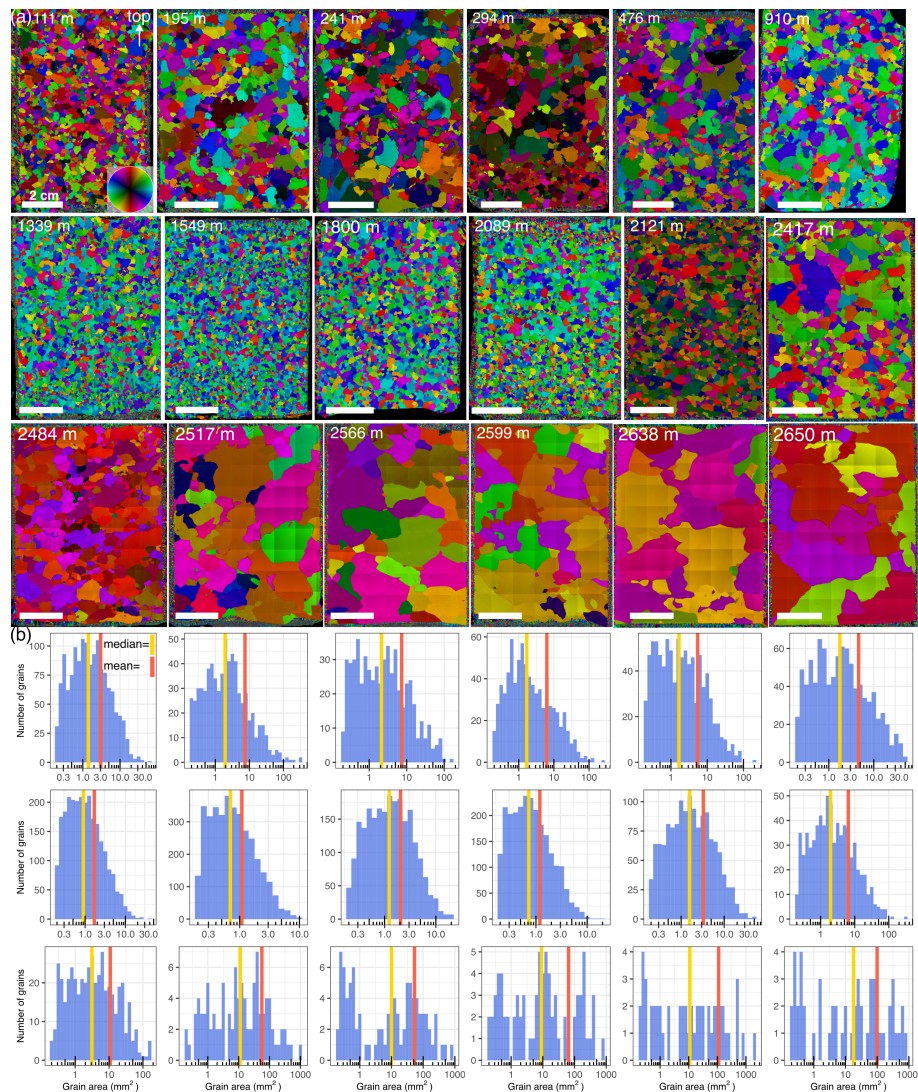

**Figure 9.** Representative selection displaying the a) microstructure and b) respective grain size distributions (note the varying axes) at similar depths as the CPOs in Figure 3. Top indicates the surface of the ice sheet. The c-axis orientation is indicated by the colour wheel.

## 3.4 Signs of disturbed stratigraphy

EastGRIP DEP, ECM, and microstructure data do not show correlating indications of abrupt disturbances until a depth of 2618
m. At 2618.3 m, a sudden change in conductivity coincides with a change in crystal size and CPO over a few centimetres (Fig. 10). Very large ice crystals (up to 7 cm in diameter) are followed by much smaller crystals and a more blurred multi-maxima CPO (Fig. 10).

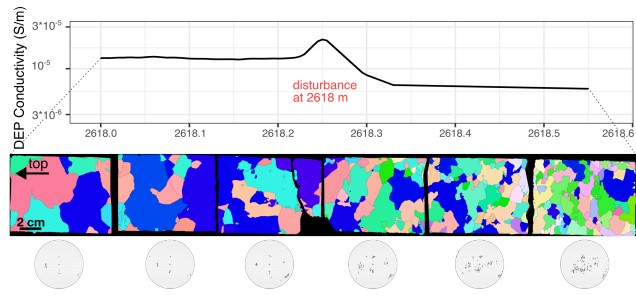

**Figure 10.** At 2618.3 m, DEP conductivity drops correlating with a strong decrease in grain size and a change in the CPO (equal area lower hemisphere projections) indicating a disturbance in the stratigraphy.

## 4 Discussion

### 4.1 Deformation regimes in NEGIS derived from CPOs

In the EastGRIP ice core, five different CPO patterns are present (Table 1). The two main processes affecting the CPO are 1) the rotation of c-axes due to deformation activated by the applied stresses (Alley, 1988; Castelnau and Duval, 1994) and 2) the growth (positive and negative) of old grains or formation of new ice crystals with different orientations due to grain boundary migration recrystallisation (e.g., De La Chapelle et al., 1998; Thorsteinsson et al., 1997). Figure 6 shows the complex CPOs with depth at EastGRIP indicated by the variety of different proportion symmetries throughout the ice sheet. To understand this

profile, we derive information on the dominant deformation regimes from the present CPO patterns as done for other deep ice cores (e.g., Kamb, 1972; Alley, 1988; Thorsteinsson et al., 1997; Wang et al., 2002; Montagnat et al., 2014a; Weikusat et al., 2017).

#### 4.1.1 Broad single maximum CPO by uniaxial compression in shallow ice

Between 111 and 196 m, we observe weak broad single-maximum CPOs. Here, most c-axes are loosely oriented towards the

vertical axis. While a sequence of complex processes takes place in firn, including but not limited to densification, metamorphism and density-crossover phenomena (e.g., Freitag et al., 2004; Hörhold et al., 2012; Fujita et al., 2014, 2016; Montagnat et al., 2020), the initial CPO at the bottom of the firn is governed by vertically orientated uniaxial compression (coaxial dominated deformation) from overlying layers (e.g., Gow and Williamson, 1976; Thorsteinsson et al., 1997; Dahl-Jensen et al., 1997). Basal planes and their orthogonal c-axes are forced to rotate towards the axis of compression, which correlates with the

ice core axis, enabling compression as displayed in deformation experiments (e.g., Azuma and Higashi, 1985; Fan et al., 2021). However, comparing laboratory deformation experiments with observations from ice cores has to be treated with caution. An additional horizontal extension component parallel to the ice stream direction is likely but indeterminable at this depth.

#### 4.1.2 Hypotheses for the origin of the crossed girdle CPO

Crossed girdle CPOs have not been described before in ice but have been in quartz and rocks containing quartz, such as schists or quartzites (e.g., Lister, 1974; Gross et al., 1978; Carreras and Garcia Celma, 1982; Law et al., 1986). Our crossed girdles are less distinct than examples described in the literature but differ from the broad single-maximum and vertical girdle above and below, respectively (Fig. 3 and 4). Crossed girdle CPOs were first categorised by Sander (1970) and later renamed into crossed girdle Type I and II (Lister, 1977). Type I is characterised by two girdles meeting at some distance from the intermediate strain axis and connected with a single girdle. In Type II, the two girdles meet at a point parallel to the intermediate strain axis (Schmid and Casey, 1986). Our observed CPOs vary between 1) both types and 2) different symmetry shapes regarding the intensity distribution and skeletal outline.

For quartz, different explanations exist for the crossed girdle CPO. As quartz is a well-known analogue for ice (e.g., Wilson, 1979, 1981) due to similar rheological (hexagonal crystal structure, forming of polycrystalline aggregates, deformation via basal slip) and optical properties (Wilson and Russell-Head, 1979), it is assumed that both materials behave roughly similarly during deformation, thus showing similar CPO patterns. However, limitations and differences exist; for example, non-basal plane deformation is well known for quartz (e.g., Baëta and Ashbee, 1969; Wang et al., 2024) but is rarely described in ice (e.g., Castelnau et al., 1996; Llorens et al., 2016b). Thus, the following possibilities for the crossed girdle should be considered with caution as further research on this CPO pattern is needed but beyond the scope of this study.

- **Deformation history - approximate plane strain under pure shear (coaxial deformation) conditions** (Lister and Hobbs, 1980; Lister and Dornsiepen, 1982). An intermediate deformation regime between plane strain and flattening would result in a transitional CPO between a small circle girdle and a Type I crossed girdle (Law et al., 1986). The asymmetry of the crossed girdles could be related to different coaxial and non-coaxial strain paths (Lister and Hobbs, 1980) or increasing strain during simple shear deformation (Garcia Celma, 1983; Bouchez and Duval, 1982; Law et al., 1986).

- **Activation of non-basal slip systems** (Hirth and Lothe, 1982; Weertman and Weertman, 1992) as observed in quartz (e.g., Toy et al., 2008) and ice cores (e.g., Fukuda et al., 1987; Hondoh et al., 1990; Shearwood and Whitworth, 1991; Weikusat et al., 2011), resulting in the fast movement of short edge dislocation segments on non-basal planes (for details see Fig. 7 in Toy et al. (2008) and e.g., Etchecopar (1977); Schmid and Casey (1986)) providing mechanisms for accommodating heterogeneous strain and the multiplication of basal dislocations, such as Frank-Read sources (Frank and Read, 1950).

- **Overprinting of an older CPO during deformation** (Christie, 1963; Llorens et al., 2022) caused by the establishment of NEGIS in its current form 2000 years ago (250 m) (Franke et al., 2022; Jansen et al., 2024).

- **Transition between broad single-maximum and vertical girdle CPO** resulting in an intermediate CPO resembling the crossed girdle CPO. Samples measured between 1064 and 1264 m of depth (Fig. 19 in Fitzpatrick et al. (2014)) from

the West Antarctic Ice Sheet (WAIS) Divide ice core display similar crossed girdle CPOs despite being retrieved from a less dynamic site.

An in-depth investigation of the a-axes of EastGRIP samples between 200 and 250 m using Electron backscatter diffraction (EBSD) is required to test the hypotheses above but is beyond the scope of this overview study.

### 4.1.3   Uniaxial horizontal extension

From 294 m downwards, the crossed girdle changes into a broad vertical girdle CPO, which becomes more distinct with depth and is a fully established vertical girdle between 493 m and 1230 m. The vertical girdle is explained by uniaxial extension further impacted by dynamic recrystallisation. During uniaxial longitudinal extension, as presumed for an ice stream, crystals rotate, and c-axes rotate away from the extension direction and, thus, the basal plane rotates towards the extensional direction. The vertical girdle plane in pole figures becomes oriented perpendicular to the axis of horizontal extension (Jacka and Maccagnan, 1984; Fujita et al., 1987; Alley, 1988; Lipenkov et al., 1989; Thorsteinsson et al., 1997; Wang et al., 2002; Llorens et al., 2022). By combining visual stratigraphy and fabric data, Westhoff et al. (2021) demonstrated that the c-axes forming the girdle are orientated orthogonal to the ice flow present at the surface, proving that extension in the ice stream flow direction is the primary driver.

Vertical girdle CPOs were observed in the Antarctic ice cores Vostok, WAIS, EDML, and Mizuho (Fujita et al., 1987; Lipenkov et al., 1989; Fitzpatrick et al., 2014; Weikusat et al., 2017), and in the NorthGRIP ice core, Greenland (Wang et al., 2002). However, the impact of this fabric pattern on internal ice deformation is difficult to assess (Wang et al., 2002; Craw et al., 2018). In a vertical girdle, some crystals will be in a soft position, i.e. with c-axes 45° from the vertical compression axes. In contrast, other crystals are in a hard position against vertical compression.

### 4.1.4   Uniaxial horizontal extension complemented by a simple-shear component

The development from a vertical girdle to a vertical girdle with a defined horizontal maxima CPO occurs between 1230 and 1394 m of depth, which is precisely the period of the transition from the Last Glacial to the Holocene (Mojtabavi et al., 2020) and is displayed in the broadening of the eigenvalue pattern (Fig. 7). At EastGRIP, the Holocene covers the upper 1240 m, followed by the Younger Dryas at 1240–1280 m, and the Bølling Allerød at 1280–1375 m (Mojtabavi et al., 2020). In contrast to NorthGRIP, the vertical girdle does not develop a maximum in the vertical direction parallel to the core axis (Wang et al., 2002), but a maximum of varying strengths in the horizontal plane (Fig. 3). Only Glacial ice displays this CPO, which is thus likely related to the rheological differences between Holocene and Glacial ice. Glacial ice has a higher number of insoluble particles, such as dust (e.g., Paterson, 1991; Stoll et al., 2021a, 2023), deposited during colder temperatures, and is characterised by smaller grains. Even though the detailed interplay between grain size and microstructure, impurities, and CPO remains ambiguous (Eichler et al., 2017, 2019; Stoll et al., 2021b), the differences between climate periods are unmistakable. The horizontal maxima of the CPO could be strengthened by an additional simple-shear component with a vertical shear plane as seen in a shear-margin core from ice stream B, West Antarctica (Jackson and Kamb, 1997) or the modelling work of Azuma

(1994). Llorens et al. (2022) model the CPO development under ice-stream flow conditions first resulting in a vertical girdle due to uniaxial extension followed by horizontal point maxima with a girdle component due to simple shear. Kamb (1972), Burg et al. (1986), and others showed the development of horizontal maxima CPOs with varying strengths in simple shear deformation experiments. Similar results were obtained with observed and modelled seismic data (Smith et al., 2017; Lutz et al., 2022). If simple shear occurs as a secondary deformation regime, in addition to the longitudinal extension, a vertical girdle CPO with a horizontal maxima would be the outcome. The comparably minor rheological differences within glacial ice, such as between stadials and interstadials or between cloudy bands and purer ice (Stoll et al., 2023), and different rates of dynamic recrystallisation due to various levels of impurity content could explain the wavy eigenvalue pattern (Fig. 7).

### 4.1.5 Dynamic recrystallisation close to bedrock

The deepest depth range (below 2500 m, temperature above -7.5°C) is characterised by multi-maxima CPO patterns of coarse-grained ice with interpenetrating grains of amoeboid shapes and curved, interlobate grain boundaries (Fig. 3, 10, 5). Multi-maxima CPOs were observed in a few polar ice cores, such as the North Greenland Eemian Ice Drilling (NEEM) (Montagnat et al., 2014a), Byrd (Gow and Williamson, 1976), Cape Folger (Thwaites et al., 1984), and Law Dome (Zichu, 1985) and in temperate glaciers and ice caps (e.g., Rigsby, 1951; Hooke and Hudleston, 1980; Tison and Hubbard, 2000; Hellmann et al., 2021; Monz et al., 2021; Disbrow-Monz et al., 2024). Like EastGRIP, the present transitions to the multi-maxima CPO are often abrupt, accompanied by a change towards larger grain size, and occur close to bedrock where temperatures are typically high. Depending on the strain rate and temperature, different recrystallisation processes impact the microstructure (Alley, 1988; Castelnau and Duval, 1994; Faria et al., 2014, e.g.,). The distinct CPOs of clustered c-axes are partly a function of the grain sampling. However, the multi-maxima CPOs, large interlocking grains, and interlobate grain boundaries are evidence of strong dynamic recrystallisation by grain-boundary migration (Gow and Williamson, 1976; Alley, 1988). The intensification of dynamic recrystallisation is fostered by the high temperature close to bedrock (above -10°C) (Fig. 1), high stresses, and large strains inflicted by potential basal shearing (Faria et al., 2014). Another possibility is a regime of stagnant ice close to bedrock which would also result in large interlocked grains as observed at the bottom of the Talos Dome ice core (Montagnat et al., 2012). Future borehole logging data will help in exploring this possibility.

The distinct microstructure in the form of large, amoeboid grains (Fig. 10b), the multi-maxima CPO pattern (Fig. 5), and the high ice temperature in the deepest 163 m (Fig. 1) enhancing grain-boundary mobility significantly indicate that nucleated migration recrystallisation is the dominant process in this depth regime. Indications of dynamic recrystallisation have been observed in EastGRIP from shallow depths downwards (Stoll et al., 2021a) similar to other ice cores (e.g., Alley, 1988; Weikusat et al., 2009; Kipfstuhl et al., 2009; Faria et al., 2014)

The multi-maxima CPO is not primarily a sampling issue caused by measuring grains multiple times as recently discussed (Monz et al., 2021; Disbrow-Monz et al., 2024). It persists when combining data from six vertically adjacent samples (Fig. 9), resulting in statistically relevant numbers of analysed crystals ($\geq$ 211). A distinct study utilising high-resolution optical microstructure data will follow up on this.

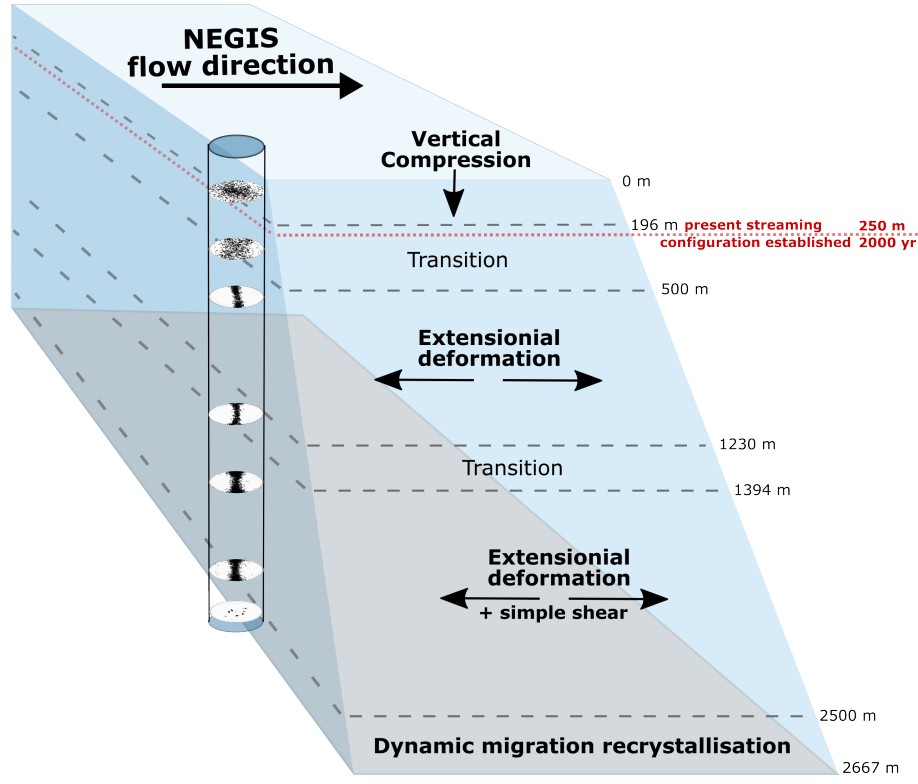

**Figure 11.** Sketch of NEGIS displaying the CPO patterns and the derived deformation regimes and recrystallisation processes. The red dotted line displays the depth when NEGIS was established in its current form (Jansen et al., 2024), everything below accumulated before today's ice stream conditions.

## 4.2 Patterns of CPO with depth compared to other ice cores

EastGRIP is the first deep ice core drilled through an ice stream and we summarise the CPO patterns and derived deformation regimes in Figure 11. To investigate differences to traditional (palaeo-climatologically motivated), less dynamic ice coring sites, such as ice domes and divides, we compare the EastGRIP CPOs with other ice cores with comparably large CPO data sets (Fig. 12). The respective CPOs are displayed in the original publications, and we have only summarised them here. The NEEM ice core was drilled on the Greenland ice divide only approximately 440 km northwest of EastGRIP, and was included due to this proximity and its similar age and temperature profiles (Montagnat et al., 2014a). NEEM is characterised by a progressive c-axis orientation strengthening with depth, resulting in a single maximum. EDML, at Kohnen Station, Antarctica, represents an ice divide with an extensional deformation regime (Weikusat et al., 2017). A broad single maximum develops into a vertical girdle with depth followed by a single maximum CPO close to bedrock. Lastly, the South Pole Ice Core (SPICE) at the South Pole, Antarctica, represents flank flow and a relatively fast surface flow velocity of 10 m/yr (Casey et al., 2014; Voigt, 2017). SPICE is characterised by a broad single maximum CPO close to the surface and a vertical girdle strengthening with depth.

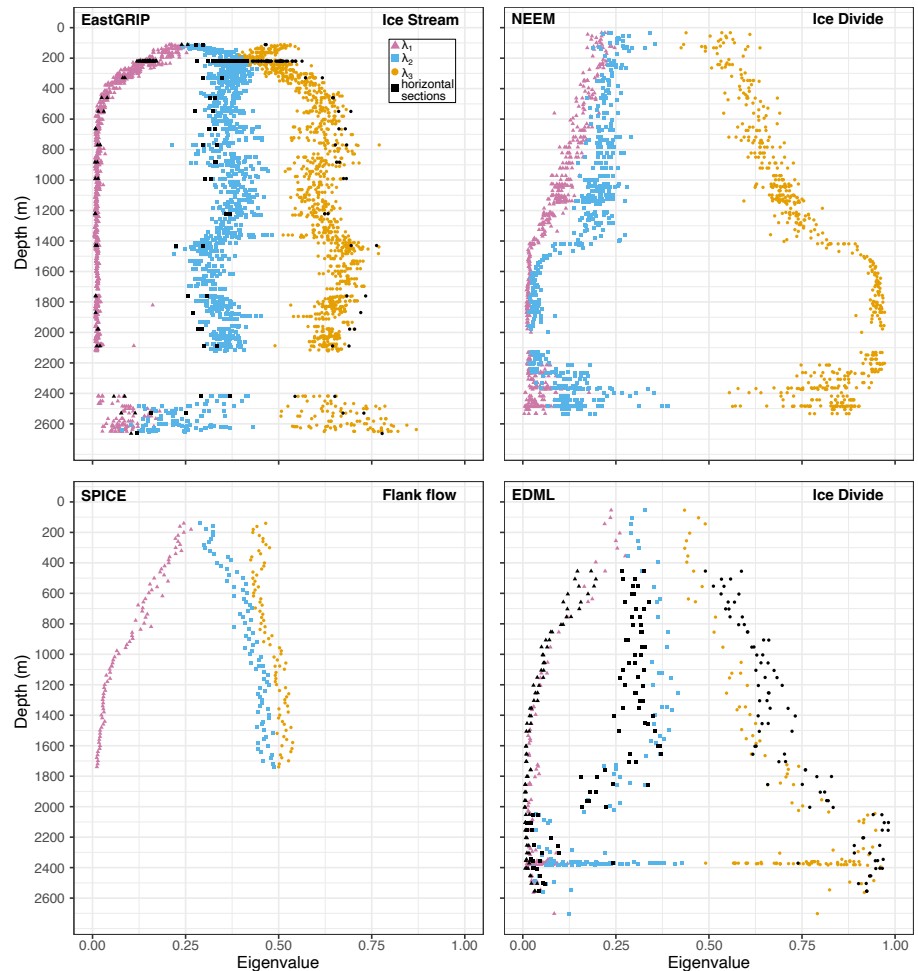

**Figure 12.** Comparison of the anisotropy between EastGRIP, NEEM, EDML and SPICE via their eigenvalue pattern with depth (Montagnat et al., 2014a; Weikusat et al., 2017; Voigt, 2017). Note the differences between maximum values of $\lambda_3$ of EastGRIP and SPICE compared to NEEM and EDML. Annotations as in Figure 7; all axes are formatted the same for better comparison.

The distinct development of a strong anisotropy with depth is characterised by the rapid decrease of $\lambda_1$ at EastGRIP compared to the other cores (Fig. 12). $\lambda_1$ stabilises slightly above 0 at 500 m depth; similar values do not occur until depths greater than 1400 m, 1500 m and 1700 m at NEEM, EDML, and SPICE, respectively. Consequently, $\lambda_3$ at EastGRIP reaches higher

values at shallower depth than in the other cores and remains between 0.5 and 0.8 until bedrock. In contrast, NEEM shows an abrupt increase in $\lambda_3$ at 1419 m, $\lambda_3$ of EDML gradually increases between 1600 and 2000 m whereas $\lambda_3$ of SPICE remains between 0.42 and 0.54. The CPOs with depth at EastGRIP further differ strongly from the other cores. In addition to the novel crossed girdle, the vertical girdle CPO at EastGRIP occurs at much shallower depth (~300 m) than at EDML (~950 m) or SPICE (~1000 m).

This comparison shows that the anisotropy development at EastGRIP is uniquely rapid. Thus, ice anisotropy plays a crucial role for ice flow within NEGIS and likely other ice streams. Even flank flow at SPICE differs strongly from the conditions at EastGRIP. The potential implications and consequences of this finding are discussed in section 4.5.1.

     No other core exhibits wavy fluctuations in eigenvalues with depth as pronounced as EastGRIP (Fig. 12). The wavy pattern correlates with the depth regime characterised by the vertical girdle CPO, but SPICE, containing a similar vertical girdle,
displays no such eigenvalue pattern. A part of the EDML eigenvalues, roughly correlating with the vertical girdle CPO, could be interpreted as a weak wavy pattern, but is indistinct compared to EastGRIP. The broader spread between 650 and 950 m correlates with the brittle zone at EastGRIP (Westhoff et al., 2022) and could be explained by the formation of small grains close to bubbles transforming into clathrates and natural variability. This difference could also be an artefact from the much larger EastGRIP data set displaying more variation. To explore this further, future studies on the single crystal scale and along
particular depth regimes are needed at specific depths, such as the brittle zone, and for example, the depths around 1370 and 1895 m (Fig. 7).

## 4.3    Confirming the hypothesized vertical girdle CPO in ice streams

     Our observational data enable us to confirm model assumptions about CPO patterns and deformation modes in active ice streams (e.g., Alley, 1988; Azuma, 1994; Llorens et al., 2022) enabling the transfer to other ice streams in Greenland and
Antarctica. These studies assume that the c-axes of ice crystals rotate away from the axis of extension and thus produce a vertical girdle CPO with a potential horizontal maxima pattern. However, so far, only observations of the ice flow velocity on the surface of NEGIS indicate extensional behaviour. We can now confirm that within NEGIS, the most abundant deformation regime is longitudinal extension represented by a vertical girdle CPO.

## 4.4    Grain-size with depth in an ice stream

The examined grain-size trends with depth in EastGRIP are comparable to those in other Greenland deep ice cores (e.g., Thorsteinsson et al., 1997; Montagnat et al., 2014a). Figure 8 displays that grain size with depth is similar in EastGRIP and NEEM. The more specific interpretation of grain size data will be presented in follow-up publications, as fabric analyser grain sizes (transmitting-light microscopy) are less accurate and generally coarser than reflected-light methods (Binder et al., 2013).

     Both cores show medium grain sizes throughout the Holocene, driven mainly by normal grain growth. In both cores, grain
size decreases over 150 m of depth at the Holocene-Last Glacial transition. However, this process starts at 1100 m and 1350 m at EastGRIP and NEEM, respectively. The transition in NEEM is comparably consistent, while EastGRIP shows more fluctuation in mean grain size, which could be due to the higher number of samples. In the Glacial, grains remain small as described for other deep ice cores (e.g., Gow and Williamson, 1976; Lipenkov et al., 1989; Weikusat et al., 2017). Roughly 200 m above bedrock, grains are strongly curved, interlope into each other, and get significantly bigger. We interpret these features
as evidence for 1) chemically purer ice formed during the warmer conditions in the Eemian period (Thorsteinsson et al., 1997) and 2) very warm ice (Fig. 1) caused by the proximity to bedrock and, thus, more geothermal heat. In the deepest part, the

mean grain size at NEEM, also obtained with an automated fabric analyser and with partly larger samples, reaches up to 922 $mm^2$, while crystals at EastGRIP only reach about 280 $mm^2$.

Despite different CPOs and dynamic conditions (EastGRIP - ice stream; NEEM - ice divide), the similar grain-size profiles between both cores suggest that grain size is not significantly impacted by fast-flowing ice stream ice. It further indicates that grain growth and grain size reduction are driven similarly. Most NEEM samples are in a similar size range as EastGRIP, even close to bedrock. On smaller scales, the interplay between grain size, impurities, and CPO is more complex and remains challenging to disentangle (e.g., Paterson, 1991; Eichler et al., 2019; Stoll et al., 2021b).

## 4.5 Implications

### 4.5.1 Relevance for ice sheets and ice-flow modelling

The presented data indicate that NEGIS, at the location of EastGRIP, mainly experiences extensional deformation along flow resulting in heterogeneous plug flow. Contrary to block flow this includes additional internal deformation components and heterogeneous strain and stress phenomena on different spatial scales indicated by the crossed girdle and vertical girdle with horizontal maxima CPOs. Despite their importance for ice-sheet and glacier mechanics, these microstructural features are usually overlooked in models (Faria et al., 2014). To estimate the overall impact further, analyses regarding the mechanic responses of a crossed girdle CPO compared to the broad vertical CPO below or isotropic ice are needed. We discuss the crossed girdle CPO for the first time in natural ice, and, to our knowledge, most ice fabric modelling has yet to produce similar CPOs. Thus, enhancing the bridging of different scales and approaches, as recently done by Richards et al. (2023) and Ranganathan and Minchew (2024), is crucial.

A fundamental question arises regarding the upstream source and flow path of ice at various depths at EastGRIP, especially when considering the significance of the vertical fabric profile for ice sheet dynamics and the modeling of NEGIS. One key aspect is whether one assumes a constant velocity field in the past, where ice particles flow into NEGIS through the shear zone. This assumption is commonly used in modeling, mass-flux, and firn compaction studies (e.g., Holschuh et al., 2019; Gerber et al., 2021; Franke et al., 2021; Oraschewski and Grinsted, 2022; Gerber et al., 2023). Contrary to this assumption is the idea that shear zones in the upstream sector of NEGIS move with the ice over time and can cause the ice stream to either widen or narrow, reflecting a highly dynamic component in the development of NEGIS (Grinsted et al., 2022; Franke et al., 2022; Jansen et al., 2024). The different consequences arising from these two perspectives are highlighted in Jansen et al. (2024), where the sheared cylindrical folds are used as passive markers for ice flow. The fold deformation patterns at the shear margins do not align with the flow lines derived from the current ice flow velocity vectors, and instead indicate that ice remained inside the shear margins during the time of shearing.

Underlying this question is a long-standing fundamental debate on whether NEGIS, in its current form, represents a constant feature throughout the Holocene (Fahnestock et al., 2001) or whether NEGIS and similar ice streams are temporary phenomena that evolve with time and can switch on and off over a few thousand years (Franke et al., 2022; Jansen et al., 2024). While a detailed discussion of this topic is beyond the scope of this article, it is highly relevant for understanding the dynamic

behaviour of the Greenland Ice Sheet. It is clear, however, that the two theories - one positing an unchanging velocity field with ice flow through the shear zone and the other proposing a changing velocity field with moving shear zones—are fundamentally incompatible.

So far, we used the measured CPO data to establish the current deformation regimes at the EastGRIP site. However, there is a debate whether or not, and under which conditions, ice fabric can be used to reconstruct deformation and flow history

(e.g., Lilien et al., 2021; Llorens et al., 2022). However, assuming that EastGRIP ice accumulated initially at the ice divide upstream of NEGIS before NEGIS was established, respective changes in flow and deformation history, i.e. changes in stress and strain conditions, could completely change or partly imprint the fabric (e.g., Craw et al., 2018; Lilien et al., 2021; Llorens et al., 2022). Llorens et al. (2022) show that the fabric at the onset of NEGIS would be preserved for a maximum of ~7 kyr. The strong dynamic recrystallisation at the bottom of the core would likely result in a much shorter duration of preservation.

Similarly, at regions with higher strain, such as the shear margin, only ~200 years of preservation duration are estimated (Llorens et al., 2022).

The crossed girdle CPO around 250 m correlates with the timing of the establishment of NEGIS and its shear margins 2000 years ago (Fig. 11) (Jansen et al., 2024). Assuming that NEGIS in its current form is 2000 years old, as explained above, results in a unique change of deformation regimes inside the EastGRIP ice core over a short time around a depth of 250 m.

Before either the establishment of NEGIS or the movement of EastGRIP ice into the ice stream, ice below 250 m of depth mainly experienced vertical compression and thus a different starting CPO than ice above 250 m, which mainly experienced a NEGIS-induced finite longitudinal strain or extension (Ramsay and Huber, 1983) of approximately +0.75 along flow, and -0.33 (shortening) perpendicular to it during the last 2000 years (strain rates calculated with MEaSUREs ice velocity data set (Joughin et al., 2010a, b)). NEEM provides an estimate of the CPO and anisotropy profile at EastGRIP before NEGIS

became established, which is represented by a weak and strong vertical c-axis maximum in shallow and kilometre-deep ice, respectively (Fig. 12) (Montagnat et al., 2014a). The change in the dynamic situation at EastGRIP 2000 years ago could result in an overprinting of the initial CPO or the activation of different slip systems as hypothesised for the crossed girdle CPO (Sect. 4.1.2). However, deformation experiments have shown that strains of 0.2 can already completely change initial CPOs (e.g. Craw et al., 2018). Further, the basal slip system is dominant under axial compression and simple shear deformation

experiments (e.g. Bouchez and Duval, 1982; Montagnat et al., 2015; Qi et al., 2019). We thus stress that further research is crucial and, for now, the activation of different slip systems has to be treated with caution as an explanation for the crossed girdle CPOs. Further clarification could be obtained by analysing the borehole deformation via repeated borehole logging runs. So far, the derived CPO patterns seem to be reliable indicators of the ongoing ice flow while displaying remnants of the recent changes in ice dynamics of NEGIS (Jansen et al., 2024).

Our results show that in ice streams such as NEGIS, and likely others in Greenland and Antarctica, we cannot assume that the upper part of the ice column (upper two-thirds as assumed for ice divides (Dansgaard and Johnsen, 1969)) is dominated by vertical compression. Our observations reveal that specific stress- and strain regimes occur and must thus be represented better in ice-flow models. The novel insights from the EastGRIP ice core could help to take the next step towards more comprehensive ice flow and fabric modelling. The strong recrystallisation close to bedrock make the detection of potential shearing at the base

very challenging. Unfortunately, ice does not contain strain markers as other rocks do, thus hampering the clear identification of changes in strain with microstructural methods. Repeated logging along the entire borehole is needed to detect and quantify these changes. However, long enough periods (often up to 12 months) between measurements are required to detect changes in the borehole geometry and, thus, potential shearing with certainty.

Our data, together with recent studies on the history and stability of NEGIS (Franke et al., 2022; Jansen et al., 2024), clearly display the complicated flow behaviour of NEGIS. These new insights imply that any changes regarding the flow behaviour of NEGIS, as the recently seen acceleration in several of its regions (Grinsted et al., 2022; Khan et al., 2022), could impact the entire thickness of the ice column resulting in higher solid ice discharge in the future if surface flow velocities keep increasing.

### 4.5.2   Potential for a record reaching back to the Eemian

Comparing EastGRIP DEP and ECM data to ECM data from the NorthGRIP ice core from Central Greenland shows a good correlation between both cores for their deepest parts (Fig. 2). We thus extend the EastGRIP age-depth relationship with certainty until 104 ka b2k (2558 m). In deeper ice, the large grain sizes (Fig. 8 and 5) and high ECM values suggest that the EastGRIP ice core contains Eemian ice similar to, e.g., NEEM and NorthGRIP (NEEM community members, 2013; Rasmussen et al., 2014). However, this remains speculative until further dating, including the stable water isotope and gas records, becomes available.

The distinct change from large to small grain size at 2618.3 m correlating with a drop in electric conductivity (Fig. 10) indicates a disturbance, potentially by shearing, of the stratigraphy at this depth, which remains to be further characterised. However, EastGRIP has the potential to enable further investigation of the climate and the conditions of the Greenlandic Ice Sheet during the last interglacial.

### 5   Conclusions

The EastGRIP ice core enables the first overview of deformation regimes and CPO development throughout an ice stream. EastGRIP is characterised by five major deformation regimes accompanied by transition zones. The shallowest regime has undergone vertical compression from overlying layers yielding a broad single-maximum CPO. Below, we describe a crossed girdle CPO for the first time in ice and provide hypotheses for its origin, which remain to be tested in future studies. The major regime is extensional deformation, which is indicated by a vertical girdle CPO. With depth, extensional deformation remains dominant but the vertical girdle is extended by horizontal maxima. Close to bedrock high temperatures prevail resulting in dynamic migration recrystallisation visible in a multi-maximum CPO and large grains and, thus, few data points hindering the detection of the dominating deformation regime and the detection of potential basal shearing. We show that anisotropy develops at considerably shallower depths than at less dynamic sites, such as ice divides and, even, ice flanks. The overall plug flow shows various small-scale variations, which require further research. Additional borehole logging is needed to decipher potential shearing above bedrock. We provide a chronology for the EastGRIP ice core to 108 ka b2k below which the records of NorthGRIP, NEEM and EastGRIP become significantly different. There are indications that the EastGRIP ice core contains

ice from the Eemian interglacial period, but we find it likely that the ice-core record is stratigraphically disturbed in the deepest section. Additional grain size and ice temperature profiles with depth contribute towards a better understanding of the rheological behaviour and flow behaviour of ice streams while providing crucial data for improving future ice-flow models.

# Appendix A

## A1    Appendix A1 - fabric analyser data correction

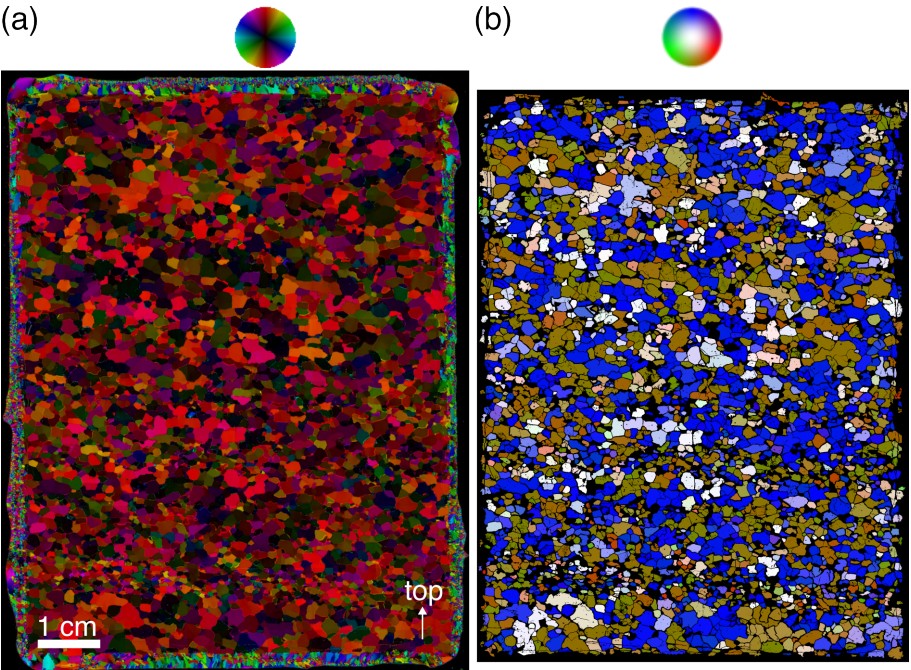

**Figure A1.** Measured fabric image and processed cAxes image from a depth of 1565.3 m. a) Original fabric image derived by the G50 fabric analyser. b) Image after processing with cAxes. The surrounding artificial ice, from glueing the sample to the plate, was digitally removed. The colours represent the orientation of the c-axis according to their respective colour wheels.

## A2 Appendix A2 - chronology

The last two tie points of Gerber et al. (2021) have been omitted as they were not in line with the match at larger depths. As observed in earlier work, most match points are located within interstadials, as the higher concentrations of alkaline dust in stadial ice neutralize any acid and mutes the ECM and DEP signals. In most interstadials, the ECM records are very similar between EastGRIP and NorthGRIP (Fig. 2), but in the long GI-23.1, only the large-scale variations between stadial and interstadial signal level are similar, and we thus have a long interstadial section without any match points. In addition, we have no clear match points within the short GI-18. In order to avoid having to interpolate over the combined GS-18, GI-18, and GS-19.1 period, we have assigned a match point within GI-18 based on broad-scale patterns in ECM and DEP. The uncertainty of this match point is larger than the normal maximum uncertainty but still improves the overall accuracy of the interpolation.

The GICC05 chronology was transferred to EastGRIP by linear interpolation of depths between the match points (Mojtabavi et al., 2020). The time scale uncertainty itself is comprised of the GICC05 maximum counting error which applies to all cores synchronized to GICC05, and the time scale-transfer uncertainty. The time scale-transfer uncertainty comes from the uncertainty of the individual match points and the interpolation approach. Most match points have low uncertainty (in the order of a few cm), but occasionally, wide peaks or ambiguous alignment means that the uncertainty in rare cases can be a few tens of cm (discussed by Mojtabavi et al. (2020)).

*Data availability.* Data will be made available via PANGAEA once the manuscript has been accepted.

*Author contributions.* The study was conceptualised by IW, DDJ, SK, JK, and NS. The manuscript was written by NS, IW, JK, KD, DJ, SR with contributions from all authors. Fabric data were collected by NS, IW, JK, JE, DJ, JW, DW, TS, TH and processed and analysed by NS, JK, KD, IW, DJ with contributions from all authors. DEP data were collected by NS, DJ, JW, SK, AS, and IW and processed by FW. GS processed the ECM data which were matched by GS and SR. SR calculated the time scale. Funding acquisition by IW, PB, FW, MD, and DDJ.

*Competing interests.* The authors declare that they have no conflict of interest.

*Acknowledgements.* We gratefully thank Steven Franke, Sebastian Hellmann, Pia Götz, Ina Kleitz, Wataru Shigeyama, Ernst-Jan Kuiper, Maddalena Bayer, Nicholas Rathmann, and Eliza Cook for their assistance in the sample preparation and measurement at EastGRIP. We thank Sonja Wahl, Florian Painer, Nils Hvidberg, Yannick Heiser, Nils F. Nymand, and Mikkel Lauritzen for contributing to DEP data. We further thank everybody involved in gathering ECM data. We acknowledge Alexander Schlemmer for his technical support in setting up the environment for remote data processing. We further thank Steven Franke and Shuji Fujita for valuable input improving the quality of the manuscript. We thank the entire EastGRIP community for logistical assistance, ice core processing, and fruitful discussions. This work was carried out as part of the Helmholtz Junior Research group "The effect of deformation mechanisms for ice sheet dynamics" (VH-NG-802). Nicolas Stoll acknowledges additional funding from the Programma di Ricerche in Artico (PRA) and the German Academic Exchange Service (DAAD, Postdoc Fellowship RESTORATION). EastGRIP is directed and organised by the Centre for Ice and Climate at the Niels Bohr Institute, University of Copenhagen. It is supported by funding agencies and institutions in Denmark (A. P. Møller Foundation, University of Copenhagen), USA (US National Science Foundation, Office of Polar Programs), Germany (Alfred Wegener Institute, Helmholtz Centre for Polar and Marine Research), Japan (National Institute of Polar Research and Arctic Challenge for Sustainability), Norway (University of Bergen and Trond Mohn Foundation), Switzerland (Swiss National Science Foundation), France (French Polar Institute Paul-Emile Victor, Institute for Geosciences and Environmental research), Canada (University of Manitoba) and China (Chinese Academy of Sciences and Beijing Normal University).

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
