# Peer review of "Linking Crystallographic Orientation and Ice Stream Dynamics: Evidence from the EastGRIP ice core"

_EGUsphere, 2024_

## Author Response (AR1)

**EGUSPHERE-2024-2653**
**EastGRIP ice core reveals the exceptional evolution of crystallographic preferred orientation throughout the Northeast Greenland Ice Stream**
**Referee Report #1**

R: This paper presents a comprehensive and detailed study of the evolution of the texture (or fabric) along the EastGRIP ice core. This paper is based on a tremendous measurement campaign that must be highlighted (more than 1200 thin sections made and measured!) Overall the data are of very good quality, and their interpretation, for most of them, are reasonable, well argued and well documented.

A few interpretations can be questioned, as detailed further in this review, and I would appreciate the authors to consider these comments prior to publication. Some interpretations are not in good agreement with previous work what is not a problem by itself, but the discrepancies are not always well argued.

Another point concerns the citations. In view of respecting the work of former colleagues (and research ethics rules) I would suggest the authors to avoid citing only review papers (e.g. Faria et al.) but also the original work cited in the review papers that are sometime more closely related with the subject matter.

Appart from that, this paper clearly deserves publication in TC and I would like to emphasize the clarity and quality of the writing.

A: We thank the reviewer for this thoughtful review and appreciate the input, which certainly helps to improve the quality of the manuscript. We agree with most comments and will make sure to include more original studies in the revised version. In the following, we will reply (in blue) to the specific comments .

Specific comments :
- the authors make use of the second order orientation tensor as a proxy of the texture evolution. Please mention that this proxy is not adapted for all types of fabric, and in particular, do not discriminate some multi-maxima fabric with isotropic ones. What about the case of the crossed-girdle fabric ? How well is it represented by the 2d order orientation tensor ? What precautions should be taken when interpreting the evolution of the eigenvalues of the 2d order orientation tensor with depth ?

A: We mention this aspect in section 2.5 ("However, eigenvalues are limited to unimodal distributions due to their inability to differentiate between certain fabric types, e.g. a narrow girdle and a multi-maxima. Thus, additional data representation such as stereographic projections is required to evaluate more complex fabric patterns.") and will strengthen this point further. Keeping this in mind is certainly a precautiona as well as providing a variety of data (see below).

As the crossed girdle has not been described before in ice, it is tricky to judge the use of different proxies – we thus present the CPO patterns, the eigenvalues, and contoured stereo plots, and will add additional microstructure maps for this CPO. This provides a robust data set to the reader.

- In the abstract and along the text a specific « crossed girdle » fabric is mentioned as an originality of the obtained measurements. Nevertheless the specificity of this fabric does not appear so obvious. Would it be possible to find specific illustration or data treatment to make it stand out better? The 241 m depth pole figure that is shown in figure 2 could well be interpreted as a wide girdle with a low anisotropy…

A: We also came across this issue and tried to solve in a good way without giving it too much space in an already rather long manuscript. In figure 8 we combine the data from 6 adjacent thin sections to more clearly visualise the fabric pattern; this turned outto be the best option. We will now include microstructure maps of this depth regime and update the figure displaying the observed crossed

girdle CPO in comparison to other studies investigating this specific CPO pattern as observed in quartz.

- I missed some figures of microstructures that would illustrate the interpretations about the mechanisms at play, such as dynamic recrystallization. I am pretty sure a few nice microstructures could be shown, for each depth interval with specific characteristics, and at least in supplementary.
 A: This is a very good idea and has also been mentioned by referee 2. We will include more microstructure images in a new figure similar to the CPO overview figure (Fig. 2).

R: - Table 1 : illustration here of my comment about the crossed girdle. A transition from crossed-girdle to vertical-girdle is mentioned but not clearly shown elsewhere.
 A: The transition is visible in the provided CPOs in the appendix and very tricky to show in a single figure as it occurs gradually. We will refer to the appendix figure more directly. We will relocate the crossed girdle figure closer to the results section and will, in parallel, enhance its description.

R: Anisotropy indexes could be added in this table, similar to the one shown in figure 3.
 A: We are not sure how and which anisotropy index shall be calculated. We already show the eigenvalues (alone and in comparison), all CPOs, the woodcock parameter, and the Point-Girdle-Random distribution and will add microstructure images. We could add the Point-Girdle-Random mean values for the derived depth regimes in Table 1, but this might be rather confusing to the reader and repetitive to the figure. Performing the PCYS (Polycrystal yield surface) analysis would be a good way to indicate the degree of anisotropy. However, this is complicated and involves microstructural modelling going beyond the scope of this overview study, but could be the objective of future research.

R: - Figure 3 : please remind somewhere how the anisotropy indexes are calculated. Is the Woodcook parameter the more adapted ? Why not use ln(a3/a2) and ln(a2/a3) that evolves in a smaller interval ?
 A: The used indexes are calculated following Vollmer (1989, 1990) and are analogous to the Woodcock calculations of $ln(a_x/a_y)$ vs $ln(a1/a2)$. The PGR plot differs in a few ways: 1) it is closed, 2) it has three instead of two end members, and 3) it is based on the differences between the eigenvalues and not their ratios. We don´t judge here which one is more adapted and only decided, that for the EGRIP ice core, the PGR plot is better to represent the dynamic CPO patterns. If required by the editor, we are happy to include a ln(a3/a2) and ln (a2/a3) plot in the appendix.

R: - Figure 6 : please increase the size in the final version.
 A: We will edit this figure to include new electrical conductivity (ECM) data providing a better depth-age estimate. In doing so, we hope to increase the size and make it more comprehensive. We understand that the figure seems comparably small in this format. However, it is the maximum size allowed by *The Cryosphere* and should appear larger in the final document. We tried different formats and there seems to be no better fit without losing information or using a full page for it, which could be an option.

R: - Part 4.1 : please cite Alley et al. 1988, or Castelnau et al. 1994 for instance to illustrate the link between rotation of c-axes and deformation !
 A: Done.

R: Idem, regarding the impact of recrystallization, De la Chapelle et al. 1998 or Thorsteinsson et al. 1997, provided already an overlook of recrystallization along ice core (maybe not the first ones still), please site them (or others) instead of Faria et al. (or on top of).
 A: Done. We edited references throughout to show more of the original research.

R: - Part 4.1.1 : I don't remember in details the experiments presented in the paper by Azuma and Higashi 1985 but one should be careful when comparing deformation-induced fabrics observed along ice cores with experimental ones since, even for some of the slowest experiments made by Jacka and co-authors, the fabric results from dynamic recrystallization that takes place already above

1 % strain, and dominate from about 10 % strain… And the DRX fabrics can be way different from the deformation ones, especially in compression.
A: This is true and should be kept in mind. We will edit the section mentioning to be careful when comparing laboratory and natural observations.

R: - Figure 8 : here is the illustration of the crossed-girdle that is not so obvious, although clearer that in figure 2. Could a representation of the microstructure help ? Well, I understand that it is not easy to find a clearer illustration.
A: During writing and editing, we shifted this figure several times to find the best position in the manuscript, which turned out to be tricky. We agree, it turns out to be better to show this figure already earlier in the manuscript and to describe the CPO in more detail as also requested by referee 2. We will shift the figure to the fitting results section, include microstructure images, and display skeleton patterns displaying the CPO more clearly.

R: - Part 4.1.2 : some of the tentative explanations lack justifications.
For instance, the comparison with quartz does not really holds since, for some temperature and deformation ranges, quartz has several slip systems of similar activity, it is not the case for ice.
About the activation of non-basal slip systems : please go back to Hondoh 2000 review work where one can read that (1) the critical resolved shear stress required to activate non-basal dislocations is way more too high regarding the level of expected stresses here, (2) it is clearly mentioned that non-basal edge dislocation segments can help activating more basal dislocations by multiplying the active basal slip planes, but can not move on long enough distances to participate to deformation. As a summary, although there exist non-basal dislocations that we observe by EBSD for instance, they are first located mainly close to grain boundary and triple junctions, where local stress can be high, in the form of subgrain boundaries but it does not mean that non-basal dislocations actively participate to deformation. As such, they participate as an accommodation mechanism that facilitates basal glide. A high enough non-basal activity is required to have non-basal glide impact on fabric.
Please go a little further in showing the limitation of such an interpretation, not to leave ambiguous information in the paper that are later re-used in other papers with no clear view on the hypotheses behind.
A: Thank you, we appreciate the in-depth discussion. We only state that quartz has been shown to work as an analogue for ice (see cited references), which of course includes that there are limitations. However, as crossed girdle have been observed predominantlyin quartz and remain a challenge to interpret in glaciology, this comparison is feasible without provoking the wrong idea especially since no other reference data from an ice stream exists so far. Interestingly, both referees have very different perspectives on the slip-system hypothesis, which displays the need for a better understanding of the occurring processes. Eventually, only additional data from e.g., EBSD will help in explaining the crossed girdle CPO fully.
We will add extra text describing the limitations aiming to prevent wrong interpretations.

R: Overall, in this part the authors could give the likelihood of each suggested mechanism.
A: Providing a likelihood of the suggested mechanisms remains challenging without further, complex analyses, such as a dedicated EBSD campaign. We hope to conduct such measurements in the future, potentially enabling a ranking of the likelihoods. We can´t rank them with a good conscience, but we will discuss the different ideas in more detail.

- Part 4.1.3 :
In tension, experimental paper by Jacka and Maccagnan, 1984, could be cited ?
A: Done.

The effect of DRX on the strength of the tension-induced CPO could also, maybe, be mentioned ?
A: Done.

R: - Part 4.1.1:
If I'm right, the CPO transition referred to in this part is not shown in figure 2.
A: We do not refer to a CPO transition in 4.1.1. In the overview Table 1, transition zones are indicated starting from 294 m. These zones are however difficult to display with single images and are thus not included in Fig. 2. We thus provide all CPOs in the supplementary material. We will clarify this issue in the text and link more strongly to the appendix, which makes the transition zones more accessible.

Similarly, line 279, figure 2 doesn't seem to show the maxima of varying strengths in the horizontal plane. Nor does the supplementary. Did I got it right? I just see a broadening of CPO from ai data.
A: We might have formulated this in a confusing way. In the supplements, the varying strengths of the point maxima in the vertical girdle are visible as well as in the PGR diagram in Fig. 3. Investigating the hundreds of CPOs from the deeper parts of the core displayed in the appendix gives a good impression of this.

R: From line 285 I find the explanation based on dynamic recrystallization very confusing. First because I don't see the necessity of evoking polygonization to explain the vertical girdle CPO. In particular since polygonization (one of the mechanism of rotation recrystallization) is known (not only in ice) to weaken the CPO, at least slightly, and clearly not to strengthen it…
Again, please also refer to De la Chapelle and Duval 1998 when mentioning rotation recrystallization since their modeling include some energy calculation that help interpreting the occurrence of different recrystallization mechanisms. Also in Montagnat and Duval 2000 did we link the grain size evolution with depth with the rotation recrystallization modeling.
A: We include the citations in the revised version. We will rewrite this part of the discussion and will leave out the discussion of recrystallisation.

R: From line 295: could DRX with various level of impurity content also be mentioned to explain these rheological differences?
A: This is certainly a possibility and now included in the discussion. Bulk impurity data from this depth are unfortunately not available, but might, eventually, help to explore this possibility further.

- Part 4.1.5:
R: line 310: at the bottom of Talos Dome ice core we very likely observe some stagnant ice (Montagnat et al. 2014), if you want to find a reference to illustrate this statement.
A: That´s a great idea, we include this citation.

R: Line 314, again Faria et al. 2014 did not initiate the dynamic recrystallization conceptual models! Please refer to the original paper (cited in Faria et al by the way).
A: Done.

R: I don't see the interest about separating SIBM-N and SIBM-O mechanisms since, discontinuous recrystallization contains both mechanisms, nucleation and grain boundary migration (owing to the reduction in stored strain energy). On top of that, nucleation is much more likely to end up with orientations different from the parent grains than grain boundary migration. This sentence is at least unclear, but maybe also not very accurate.
A: We here follow the nomenclature by Faria et al. (2014b) to assist the broader TC readership in seeing the differences of the processes. We will reevaluate this sentence.

- Part 4.2: line 340 "we show that assumptions valid for other,…" what assumptions are you referring to?

A: This is indeed unclear. This sentence should display that the strong anisotropy at EGRIP is very different to other ice coring sites, which has to be kept in mind in future studies. We edit this sentence.

- Part 4.4: line 365 "no strong grain-size dependence of the dominating deformation regimes". To my point of view it also suggests that grain growth and grain size reduction are driven similarly, what goes in favor of continuous DRX, at least in the first part of the core.
A: We agree, that seems to be the case for the first part of the core. We will add this to the sentence.

- Part 4.5: First paragraph: It would be interesting (necessary?) to discuss the relative viscoplastic anisotropy resulting from these different fabrics observed along EastGRIP. In particular, is the crossed-girdle fabric resulting into a mechanical response strongly different from an isotropic fabric, or from the slight cluster fabric that is observed just above? If no, then why bother to try to simulate such a fabric development in an ice flow model?
A: That is true, the impact of this CPO pattern needs further research. As mentioned by reviewer 2, this CPO has neither been reproduced in the lab highlighting the need for further in-depth investigations. We here provide an overview about the data from the EGRIP drilling, the impact of this will certainly be explored in future studies and goes beyond the scope of this manuscript. It would be interesting to include this in ice flow models, but we see that this might be difficult. However, we encourage the community to use this observational data set in future approaches.

Lines 402-404: when discontinuous dynamic recrystallization dominates, the fabric only reflects the stress conditions, and looses the strain history, so the estimated preserved duration could be even lower in the bottom part of the core. Maybe it would be worth mentioning it.
A: Good point, we will mention this aspect.

Line 415: Please see my comment for part 4.1.2: the actual knowledge of dislocation slip systems activity and activation and the too weak information make this hypothesis of secondary slip system activity in EastGRIP very unlikely, and one of the less likely hypotheses over the ones mentioned in part 4.1.2. By bringing it back in the discussion part that way you put a relatively high weight on it and this is misleading.
At minimum, an estimation of the level of stress required to activate a secondary slip system with a high enough level of activity to explain the crossed-girdle texture must be provided. I doubt that this level of stress can be achieve at EastGRIP.
A: Additional EBSD data is required to investigate this hypothesis further. Not mentioning it at all in the discussion is also tricky, as it was found in modelling experiments. E.g., Llorens et al (2016) (Figure 10 -experiment 25) found that under compression (pure shear) and a high DRX rate, the pyramidal is even more active than the basal slip system, at aproximately 55% of shortening. The critical resolved shear stress for the basal is 50 times lower than the pyramidal, but the microstructure activates the non-basal activity due the lack of localisation and was related with a a-axis maximum. We will edit this section and include the uncertainty regarding the slip systems.

Lines 424-425: Why would shear occur DUE TO a strong recrystallization?? I would expect shear near the bedrock to induce high level of strain that, together with the high temperature, favor discontinuous dynamic recrystallization and therefore large grain size. Shear-induced fabrics are similar in deformation and recrystallization (at least for high level of strain) and should lead to a strong cluster. So either this cluster is hidden by the large grain size (too few measured orientations) or the main stress component at the bottom is not shear??
A: Our formulation was not clear enough here. We do not want to state that shear occurs due to strong recrystallization, but that strong recrystallization hampers the analysis (large grains, few data points etc) and thus a clear identification if shearing takes place. We will rewrite this sentence.

References
LLORENS, M.-G., GRIERA, A., BONS, P. D., ROESSIGER, J., LEBENSOHN, R., EVANS, L., & WEIKUSAT, I.

(2016). Dynamic recrystallisation of ice aggregates during co-axial viscoplastic deformation: a numerical approach. *Journal of Glaciology*, *62*(232), 359–377. doi:10.1017/jog.2016.28

Review of Stoll et al: "EastGRIP ice core reveals the exceptional evolution of crystallographic preferred orientation throughout the Northeast Greenland Ice Stream"
Review by Dave Prior, University of Otago.

R: I'd like to congratulate the team on collecting an astounding data set. This is a unique piece of work that must be published. This is the first whole ice sheet thickness core that that has been collected to address issues of ice dynamics, rather than palaeoclimate. The density of CPO data together with the restoration of azimuthal orientations blows all previous core studies of ice CPO off the chart. The work gives us new insights into the processes, kinematics and conditions inside an ice stream. Since flow in ice streams dominates drainage of the Antarctic and Greenland ice sheets this work is extremely important. I felt a great sense of excitement in reading the paper.

Although unique and new - the paper can be improved significantly. It needs major revisions to have the impact that it deserves and to ensure this remains a definitive description of the internal structure of an ice stream. Most significantly it lacks any comprehensive microstructural data or descriptions. I would like the authors to consider spending significant time and effort in revising and modifying this paper to ensure it is as good as it possibly can be. Better to take several more months on it rather than to rush it out close to current form. I have tried to summarise significant issues in this review document and have made a list of some of the minor things at the end. In addition, I have an annotated version of the manuscript that will be better for seeing a thorough set of minor comments and also shows some of my un-filtered thoughts on reading. Apologies for the handwriting on this - most was done on two cramped and bumpy C130 flights to and from the ice! If there are comments that the authors cannot read and want interpreted then they can get in touch. My review here reached a cut-off time, rather than me completing everything I could write!

I don't think the paper contains too much material - indeed some of my recommendations will involve putting more stuff in. The writing is verbose and the current text can be reduced by 20 - 30% simply by improving the tightness of the writing.

A: We thank the reviewer for this in-depth review and are very happy to hear the positive remarks regarding the work done over the last years at EGRIP. We further appreciate the vast input, information, and numerous suggestions. The review report will help strongly increasing the quality of the manuscript and we tried to incorporate as much as possible. We agree that more microstructure figures will improve the manuscript and will thus add more data from the fabric analyser. However, there is a companion paper in preparation utilising high-resolution Large Area Scanning Macroscope (LASM) data. This enables a higher level of detail to describe and interpret the microstructure – qualitatively and quantitatively. Due to the unmatched large size of the gathered data set, this will be published separately.
We will reply to the specific comments in blue. We will include the hand-written changes in a revised manuscript version, which will be easier to follow in a track-changes document than listing them all here.

**Major comments:**
R: Evolution? The paper uses the word evolution a lot - it is there in the title. I think this is

highly misleading. The paper presents the pattern/sequence of CPO (and other stuff) as a function of depth. There may be ice core scenarios where the sequence with depth is a proxy for time and can be inferred to be an evolution; the upper part of ice domes/ divides for example. If there is any scenario where the pattern/sequence of CPO with depth is not realistically a proxy for time or strain, it is here in an ice stream. Inferring evolution later on in the paper is important - but it does not fall out simply from the vertical sequence, it requires much more information including comparison with Greenland ice divide core data and discussion of the effects of ice transport through the shear margin. Please remove the words evolution (and other phrases that imply the same) in describing the sequence with depth. Remove "evolution" from the title. How about a title such as: "New insights into ice stream dynamics from crystallographic preferred orientations through the EastGRIP ice core." ?????

A: We see the point that "evolution" can be interpreted wrongly and thus delete it throughout the text. It was not our intention to suggest a continuous development of CPO in the vertical ice column but simply the change in the vertical CPO profile. We will also edit the title accordingly.

Discussing the shear margins, and especially the ice transport there, in more details would open up another disputed topic where no clear consent has yet been achieved inside the EGRIP community. We are currently working on microstructure and fabric data from the southern shear margin, aiming to understand better the ice flow behaviour and history inside the NEGIS shear margin. Considering the flow lines leading to the EastGRIP drill site, the shear margins, however important for the deformation of NEGIS, should not have a big impact on the CPO observed in the ice core. Exploring this further is beyond the scope of this study, and we will not discuss ice transport through the shear margins further than already done.

**R: Microstructures?** The microstructures of the ice should go hand in hand with CPO data. Both are needed for the most robust interpretation of processes and history. Holding back basic microstructural information for another paper makes no sense as CPOs and microstructures are both needed here. The microstructural data presented is very limited - just two micrograph figures from the deeper part of the core and a graph of mean grain sizes.

A: As pointed out by the reviewer, we present a huge data set which comes along with new challenges. We will discuss the higher-resolution microstructure data derived with the LASM in a quantitative way in a companion study. For this study, we will add more microstructure data from the Fabric Analyser enabling a more holistic insight.

• Figure 2 needs to have a fabric analyser microstructure picture to go with each CPO (or min a parallel extra fig). Please be careful in the colour scheme chosen for these, to maximise the comparability of microstructures, particularly within samples with similar CPOs (e.g. 1910 to 2417m). The colour scheme used in figure 1 of (Stoll et al., 2021) is not good as the colours in each image change radically depending on the (arbitrary) azimuth of the girdle. A much better colour scheme would be to colour the pixels based on the inclination: the angle between the c-axis and the core axis. The only complication then is whether grain shapes depend on orientation of the section relative to the girdle. This can be dealt with by marking ticks on the outside of the c-axis stereonet to show where the section plane is.

I would also, for completeness, have a supp info document with the microstructural image of each sample, to complement the CPO file.

A: Producing a good overview figure is challenging and we thus decided to only show CPOs here. Adding a microstructure image to each displayed sample would make each panel very small and would thus defeat the purpose of the microstructural data. We will thus add a second figure displaying the respective microstructure.

Having different colours is unavoidable as retrieving the true orientation of the core during drilling is still not possible. The used colour scheme is designed to portray the girdle CPO easily by having the same colours on "opposite sides". All data will be available upon publication enabling specific plotting approached depending on the desired analysis.

R: • In using grain size data to infer processes, grain size distributions are very useful. The means or medians alone are insufficient. I see that some distributions (down to 1340m) are shown in fig 3 of (Stoll et al., 2021). These show slightly skewed normal distributions (difficult to tell as grain area plotted on a log scale ~ same as plo[ng grain diameter on a linear scale. I presume the "counts" axis is number of grains not cumulative area?). I'd really like to see the pattern of grain size distributions over a greater depth and how they compare with GSDs from the lowermost section with multimaxima CPOs. I realise the latter may be complicated as the data sets will be small (in terms of grain number). A figure that parallels fig 2 and has a GSD for each sample from the FA data would be great. That would leave open the possibility of having a paper later on focused on the sublimation image data.

A: We discussed this internally several times and decided to focus on the CPO data in this manuscript. GSD data is without a doubt interesting, but more accurate data, especially for small grains, will be available once the full LASM data set is available. We discuss the statistical difficulties in the deepest section due to the small number of grains and thus refrained from further interpretation here. We could provide GSD plots for several depths but would prefer not to due to the mentioned reasons.

The displayed counts in Stoll et al., 2021 refer to the number of grains.

• Grain shape data - bubble shape data?
A: Detailed grain and bubble shape data analysis is beyond the scope of this study. This will be discussed in-depth and quantitatively in the companion publication (see above).

**Inference of processes.**
The inference of processes from the observations is very poor. It's not necessarily that I disagree with the interpretations, more that there is not a clear separation of observations and interpretations and ojen the text jumps to the interpretation without a useful justification. In particular you should remove expressions such as "dynamic recrystallisation has been observed…" (line 312): it is not observed, it is an interpretation. Inferring that there has been "dynamic recrystallisation" in ice is not particularly useful. All terrestrial ice deforms at very high homologous temperature. If it has deformed it will have undergone dynamic recrystallisation, so that term by itself does not say anything useful about inferred processes. Using observations to infer much more specific processes such as recovery, subgrain rotation, subgrain rotation recrystallisation, grain boundary migration, nucleation, grain boundary sliding – see for example table 1 in (Trimby et al., 1998).

A: We agree that the wording should be more precise; it will be changed in the revised version. We do not stress the role of dynamic recrystallisation in the text, but not mentioning it would be wrong either. Inferring more about the specific processes is desirable and will be done to a larger extent in the companion paper due to the higher-quality microstructure data derivable with LASM. Still, following the next comment we will enhance the text regarding the inference of processes in a dedicated subsection.

The inferences need to lean in more heavily into the results from experiments (physical rather than numerical), where often we have time or strain series microstructural data and mechanical data that allow the interpretation of process to be more robust. Comparison of

microstructural observations in samples that represent a point in time/strain with those developed as a function of time/strain then provides a good basis for interpretation. In particular, a lot of our inference comes from experiments where microstructural change is observed (transparent polycrystal, metals in an electron microscope etc) see for example (Urai et al., 1986) Here's my little checklist of common relationships between observations and reasonable interpretations. I've put some references where I think you may be unaware here but have not referenced completely.

• Strong CPO: dislocation glide and climb.
• Subgrain boundaries: dislocation glide and climb, recovery
• Cell like subgrains: recovery, subgrain rotation (particularly if evidence of a range of subgrain misorientations)- sometimes called polygonization. (Guillope and Poirier, 1979; Poirier and Guillope, 1979; Poirier and Nicolas, 1975)
• Cell like grains adjacent to subgrains of a similar size: subgrain rotation recrystallisation. (refs as above)
• Irregular-lobate grain boundaries: strain induced grain boundary migration (SIGBM). (Bestmann et al., 2005; Jessell, 1986)
• Highly irregular grains with strongly skewed grain size distributions: SIGBM with a nucleation process (subgrain rotation recrystallisation and/or bulging).
• Grains that are consistently smaller than subgrains: bulge nucleation associated with subgrain rotation. (Halfpenny et al., 2006)
• Polygonal grains with straight-slightly curved boundaries: absence of SIGBM. Normal grain growth(?), possibly associated with mechanisms involving grain boundary sliding. Grain size distributions likely to be normal or only slightly skewed.
• Polygonal grains and independent evidence of high strain: mechanisms involving grain boundary sliding.
• Jump in boundary misorientations between subgrain boundaries and grain boundaries (needs EBSD data): mechanisms involving grain boundary sliding (Bestmann and Prior, 2003; Craw et al., 2018; Fan et al., 2020), also change of miso axes from crystal control to random.
• Linear alignment of multiple grain boundaries: mechanisms involving grain boundary sliding. (Ree, 1994)

A: We appreciate the effort in sharing this list and incorporate some in the text. However, as visible with this list, discussing these processes requires a thorough (quantitative) investigation of the microstructure alone which is beyond the intended scope of thisstudy. The dedicated study on microstructural properties incorporates many of these thoughts and will be more suitable to display and interpret them in detail. We will make sure to follow the publications on physical experiments that you kindly provided for the next publication.

**Eigenvector information, especially inclination.**
The paper misses important information about the eigenvectors. This includes both the orientation of the eigenvectors relative to the CPOs and within the core reference frame. Things you should do:
• Mark the eigenvectors on the stereonets on figure 2. Red symbols would distinguish them nicely from the data points. I think eigenvectors should also be added to the stereonets in the supp info file. Given the code to generate the c-axis plots should be capable of generating these, it should not be a big deal to do. I would also consider including an error small circle or ellipse where this error is significant (error > than ~5 degrees?). This will do two really useful things. In the vertical girdle samples this will give an idea as to how well the horizontal maximum (within the girdle) is defined. In the multimaxima samples it will help illustrate the issue of the small number of grains in the

data set. I understand that error circles may be a little complex and may be beyond scope- particularly for all samples in supp info, but if you can build it into the plotting code, I would add these.

A: We will provide more information on the eigenvectors and will implemen them in the plots. For the revised version, we will try and then decide if we including errors. Calculating a meaningful error is not trivial and it could decrease the readability of the plots and does not provide significant new insights. The issue of small numbers of grains in the deepest samples is evident and already discussed in the text.

• Provide a plot of eigenvector inclination to go with the eigenvector magnitudes in figure 4. I predict that this will have l3 vertical to ~200m, horizontal from ~600 to ~2400 (highlighting the horizontal maximum within the girdle- and departures from this showing where it gets weaker) and steep (with a fair bit of variation) at >~2400m. Plot inclinations of all three eigenvectors to reveal the complete pattern. The most crucial place to use these data would be in comparing with other ice cores in fig 10. I presume you have access to the NEEM and EDML data and can generate the eigenvector inclination pattern with depth. SPICE data are all available as text files and generating the inclinations should be trivial. The eigenvector inclination data for EastGRIP will be starkly different to all of these others. NEEM will have l3 ~ vertical at all depths. I think this will be true in EDML as well: it looks like the girdle CPOs in fig 2m of (Weikusat et al., 2017a) have maxima in the girdle that are steep to vertical? The girdle section for EDML in fig 10 also has a wavy pattern (like EastGRIP: although not recognized in this manuscript I am reviewing), and the orientation of the max (l3) within the girdle helps distinguish the EastGRIP and EDML patterns. SPICE is an interesting comparison - this is a vertical girdle dominated core- the pattern in fig 10 suggests to me that whilst l1 will be horizontal at all depths (as I'd predict for the vertical girdle section of EastGRIP) there will not be a consistent pattern in the inclinations of l3 and l2 (different to what I'd predict for EastGRIP).

A: We will incorporate the eigenvectors for EGRIP in a meaningful way. However, doing this for all other cores is not trivial and would benefit from a dedicated study on its own including other ice cores than the ones we investigate here. The observed eigenvalue patterns are already very distinguishable and display the differences between e.g., EGRIP and EDML.

**Description of CPOs**

The descriptions of the CPOs is not good. A few of specific things:

• Descriptions of the vertical girdle CPOs describes them as having two maxima. This is incorrect. Clustering at points on the primitive circle 180 degrees apart are not 2 maxima – this is a single maximum and the spatial separation of clusters simply a function of reference frame choice: in this case the maximum is around a horizontal axis and the choses stereonet reference frame has a horizontal primitive circle. A reference frame with the primitive in the vertical plane paerpendicular to flow with have one spot on increased point density on the net.

A: This is correct and imprecise wording on our side. We used the two maxima wording to better display the observed CPO to a *TC* reader not familiar with stereoplots etc. We will change this in the revised manuscript.

• The word asymmetric is used. I'm unsure what this relates to. Describing patterns on stereonets as symmetric or asymmetric is uninformative. Better is to describe the nature of the symmetry and the reference frames to which those statements refer . For CPOs associated with deformation it is commonly possible to describe CPOs as having

orthorhombic (three orthogonal mirror planes) or monoclinic (A two-fold axis which is also the pole to a mirror plane) symmetry. If the internal symmetry elements (rotation axes, mirror planes) have the same orientations as symmetry elements of the external deformation kinematics, that gives a good reason to suggest that the CPOs relate to that deformation. Simple shear is a really great example: the two-fold axis of the monoclinic shear kinematics maps onto the 2 fold axis of monoclinic CPOs (Bouchez and Duval, 1982; Journaux et al., 2019; Qi et al., 2019). In your case see section "Kinematic significance of vertical girdles"

A: We use asymmetry twice, once for describing the shape of the crossed girdle "leg" outlines, and once to point out the varying strength of the single maximum within the vertical girdle as described above. As these results are neither modelled nor from symmetrically conducted deformation experiments, we think it is important to outline that the spatial distribution is asymmetric. We will edit the text to clarify this.

• Skeletons. You mention skeletal outlines, without ever using them or explaining. CPO skeletons are in common use in the structural geology community, particularly but not exclusively for quartz. They are very useful and I think you should incorporate some. Fig 6 of (Toy et al., 2008) has a lot of examples, including a variety of x-girdles. Skeletons to represent the key CPO types in fig 2 and/or as a column in table 1 gives you a useful shorthand for the CPO patterns that you can reuse in later analysis of interpretation. Skeletons of typical x-girdles will help significantly the description of these.

A: We agree and had included outlines in earlier drafts, but refrained to use them to avoid biasing the reader as this might already imply an interpretation. We will put them back in an updated figure.

• Contoured data. I am glad that you present point data - many workers do not do this and the papers are poorer for it. I think you should also have contoured data - point and contour data are complementary. In particular, it is not so easy to see the horizontal maximum in some of the point data plots and there is no real sense of the intensity of this maximum. Contoured data would help the reader resolve these features.

A: Contour data plots will be available as a supplement. However, depending on the chosen visualization, they can easily bias the eye. We thus only show them in special cases, such as the crossed girdle and the multimaxima CPOs where we incorporate several samples into one plot.

• X-girdles. As you say these have never been described in natural ice. I don't think that they have been described from experiments either. So they deserve a complete description. The one you provide is very superficial (almost nonexistent) - you jump straight to the quartz literature. The x-girdles need to be described in the observations section with skeletons to help clarity, before any link is made to qtz data. Pick several of the range of examples for description. 375_1, 375_3, 432_1, 436_1,438_6,486_1 are a suite of examples that could be used for this description, in an extra figure focused on x-girdles (with skeletons, point data and contoured figs). I am very intrigued as to what the x-girdles might mean. I think you can be a bit more discriminatory about the possible explanations. The quartz kinematic explanations (see summaries in (Law, 1990; Schmid and Casey, 1986) nearly all involve different slip systems (see fig 7 in (Toy et al., 2008). Quartz has extreme c axis patterns, for example the y-max in shear (e.g. (Cross et al., 2017) that cannot relate to basal dislocations. Although there is evidence in subgrain misorientations for non-basal dislocations in ice (Chauve et al., 2017; Fan et al., 2022; Weikusat et al., 2017b), I don't think that there is evidence from experiments that we

can get CPOs that have significant involvement of anything other than basal plane dislocations. Axial compression (Jacka and Maccagnan, 1984; Montagnat et al., 2015), simple shear (Bouchez and Duval, 1982; Journaux et al., 2019; Qi et al., 2019) are all explicable in terms of rotation on the basal plane plus preferred growth of grains with high resolved shear stress on the basal plane.

A: This is true. We did not want to focus too much on them due to their not (yet) fully resolvable origin. In earlier drafts, we included a more detailed interpretation but decided to compress it into shorter points. Both referees have very different perspectives on the slip-system hypothesis, which displays the need for a better understanding of the occurring processes.

We will describe the CPO in more detail in the results section including an updated figure.

• You say that you have no clear evidence of overprinting; but what does an overprinted CPO pattern look like? For most crystalline materials at high temperature there is very little obvious evidence that old CPOs are preserved through significant strain. Some experiments in ice (Craw et al., 2018) show that a pre-existing girdle CPO can be annihilated with only 20% strain. This does not mean I don't believe we can get CPOs with some overprinting I suspect it means that they have low chance of survival.

A: We agree that it is unlikely for CPOs to survive and to be recognized as overprinted CPOs. We will implement the mentioned aspects and edit this section.

**Kinematic significance of vertical girdles**

I am really intrigued by girdle CPOs. These have not been reported from laboratory deformation experiments, although there are very few experiments in axial extension and even fewer that have CPO data. Warm axial extension experiments (-3C: (Jacka and Maccagnan, 1984), -5.5C (Craw, 2016)) give an open cone CPO indistinguishable from that generated in axial compression experiments at similar temperatures (e.g.(Jacka and Maccagnan, 1984; Montagnat et al., 2015)) with a difference in shape preferred orientation (parallel to the CPO cone axis for extension and perpendicular for compression). There are no lower temperature extension experiments I know of. Although you use the beautiful study from Fujita et al (シュウジ, 1987) as a justification of these being related to extension, I think you can and should do more to develop this idea from your own observations. Some of the kinematic context of the site is only outlined at the end (line 411-412). This provides a framework for understanding that should be earlier and more detailed. You can then relate the orthorhombic internal symmetry of your vertical girdle CPOs to the orthorhombic symmetry of the ice stream extensional reference frame, making the idea that these are related to axial extension really robust.

A: Vertical girdle CPOs were observed in natural samples from e.g., ice divides and have been interpreted as indicators of extension in several studies (such as Lipenkov et al 1989, Thorsteinsson 1997 and others cited in the text). The CPO pattern was also predicted numerically (e.g., figures 4, 5 and 6 in Llorens et al., 2022). The connection between the kinematics and the vertical girdle CPO has thus already been hypothesized and is now confirmed.

It would be great to get more laboratory data from experiments conducted at ice sheet temperatures aiming to retrieve vertical girdle CPOS.

These CPOs give me as an experimentalist a big challenge. I cannot explain them using the same ideas we use to explain axial compression and shear CPOs (Qi et al., 2017; Qi et al., 2019; Wang et al., 2024). At low stress natural conditions we expect the recrystallisation

effect, giving orientations with high resolved shear stresses on the basal plane, to dominate. This would be an open cone rather than a girdle.
Very puzzling.
A: Indeed, the CPOs are fascinating. It would be great to see laboratory experiments aiming to recreate these CPOs as they now have been observed in natural samples and in modelling exercises.

R: Related to this point I would suggest that plotting Fig 2 could be plotted in a kinematic rather
than geographic reference frame. Or at least make the kinematic reference frame clear within the geographic frame of this figure.
A: As explained above, we do not have the true orientation of the samples and assuming a kinematic reference frame (especially with depth) is too much speculation. We refer to Westhoff et al. 2021 where the broader kinematics in reference to dedicated CPOs is discussed, but this can´t be extended to the entire core.

**Strain terminology**
The "strain" described in Line 411 is not precise. Stretch and strain are not the same and I get different implications if I take these (175%, 67%) both as stretch, both as natural strain or one as stretch and one as strain. Tighten up the language (ask Paul)
A: We will change the text to "Before either the establishment of NEGIS or the movement of EastGRIP ice into the ice stream, ice below 250 m of depth mainly experienced vertical compression and thus a different starting CPO than ice above 250 m, which mainly experienced a NEGIS-induced finite longitudinal strain or extension (Ramsay and Huber, 1983) of approximately +0.75 along flow, and -0.33 (shortening) perpendicular to it during the last 2000 years (strain rates calculated with MEaSUREs ice velocity data set (Joughin et al., 2010a, b)".

**Kinematics, conditions and mechanisms.**
The interpretive section is a bit mixed up. Key things to infer from CPOs and microstructure are kinematics, conditions and mechanisms. These are rather intermixed at the moment; for example title for 4.1.3. is kinematic, 4.1.5. is process (and not v useful). It would give much greater clarity to separate these three issues. Kinematics will come primarily from the CPOs and their relation to the ice core setting. Grain and bubble shapes could be important here as well and are given little mention. Mechanism interpretations come primarily from microstructures and a little from CPOs. The microstructural description needs beefing up as mentioned earlier. Some conditions (T for example) are available from borehole measurements others (stress magnitude, principal stress orientations- note this is not exactly the same as kinematics) need some discussion that involves data from the ice.
A: As described above, a full description and analysis of the microstructure is work in progress and beyond the scope of this study. We will add more microstructure images and description. However, we still think it makes more sense to discuss the derived depth regimes in the current way even if kinematics, conditions, and mechanisms are not always fully separated (which they are neither in nature). This is appropriate for such a large study focusing on an overview, which could be followed by smaller studies focusing on the different components.

**Changes at the ~glacial?**
Discussion related to fig 5 seems oversimplified. The grain size changes at "glacial" are considered to be very similar- I disagree and think that this is important. The mean grain size as a function of depth at NEEM does have an abrupt step that corresponds to the depth marked as NEEM Glacial. The pattern of grain size in EastGRIP is a smooth transition that

does not have an abrupt change at the depth marked as EastGRIP Glacial. Add to this that NEEM has an abrupt and substantial CPO change (from weak girdle to strong vertical max) at the depth marked as NEEM Glacial, whereas EastGRIP is a subtle change involving a gradual increase in strength of the horizontal maximum within the vertical girdle. If you scan through from 1000m to 2000m in the supplementary data there is no obvious change in CPO visible in the EastGRIP point data stereonets. You suggest that NEEM might represent a starting profile which has evolved in the ice stream to the EastGRIP profile. Maybe the Glacial is not so important in terms of processes and kinematics in EastGRIP . So the processes are trying to end up with the same kind of ice, but the pathway is different depending of the starting material: different above and below the Glacial depth.

A: We do not agree with this observation. NEEM grain size at the glacial transition does not change rapidly but over ~150 m, which we would not call a "step". It seems steeper than at EGRIP, but not so if grain size at the exact EGRIP glacial depth is compared to the next (deeper) bag. Grains from the early Holocene, Younger Dryas, and Bolling Allerod (1150-1350 m) are smaller than at comparable NEEM depths. Ongoing work tackles grain size development in detail utilising a model using a composite flow law based on Kuiper et al. 2020a,b potentially delivering further insights into the processes at work.
We will edit this section and deepen the discussion in the revised version.

**Signature changes in fig 6.**
A lot is made concerning the change in grain size at ~ 2618m. There are a large number of similar steps or perturbations in the EastGRIP DEP profile in fig 6a. Some of these have significant changes in CPO:
• 2417-2419 transition from vertical girdle to multimax
• Between 2419 and 2450 change from multimax to vertical girdle
• 2450-2499 complicated (and interesting) CPO transition.
I don't understand the big deal with that grain size change when there are radical changes in CPO in the same sequence. What are the microstructural transitions (if any) corresponding to these CPO transitions?

A: The sudden change in grain size is the most stunning issue here implying a change of the climate period. There are more peaks in the DEP profile, but here several observations line up enabling a (rough) age-depth characterization. It is beneficiary to the community to include this climate/dating relevant information here enhancing the age-depth relationship in NE Greenland and in an ice stream. Few people expected NEGIS ice to be comparably old – now it´s one of the few cores in Greenland containing Eemian ice. To strengthen this point, we aim to include new Electrical Conductivity Measurement (ECM) data supporting our age estimates. Providing this data is fitting for the broader *TC* audience and displays the usefulness of combining different research fields within the EGRIP project and ice core research in general.

**Cartoon at the end.**
The final discussion at the end of the paper is very hard to follow and really needs a cartoon to explain it. I would have in the cartoon:
    1.   A simplified representation of key features of the EastGRIP core. Fig 7 has the CPO represented very nicely that could form the basis of this part of the figure. Fig 7 is too early in the paper . In terms of representing key aspects of the core it does not summarise the microstructure.
2.         A simplified representation of key features of an ice core that represents ice before it becomes involved in ice stream. This should highlight the same CPO and microstructural characteristics as 1. This could be an actual ice core (NEEM? – as is used in discussion), or

could be a synthesis of features from several cores (NEEM, NGRIP, GRIP, GISP2).

3.        Stratigraphic tie lines- lines that link points in depth in the two core cartoons.

4.        For some specific depths, a series of cartoon sketches suggesting how CPO and microstructure has changed as a function of time / strain. You could also have schematic graphs of how eigenvector ratios and orientations change as a function of time/ strain.

5.        Some representation of what might happen closer to the shear margins and to ice passing through the shear margin and what influence this might have on features in the EastGRIP core.

A: This is an interesting idea. We will notrepresent the full microstructure due to the reasons explained above, an updated version of figure 7 with the microstructure could be part of the companion paper. We will move figure 7 to a later position, even though it makes sense to see it as soon as the discussion regarding the regimes begins.

This is the first ice core from an ice stream, and much information regarding the broader context, such as shear margin behaviour, a complete depth-age relationship, and borehole logging, is still missing. It is still too early to draw a cartoon trying to explain everything. Figure 10 displays the eigenvalue evolution of NEEM; it would thus be redundant to show this again. Furthermore, we have already mentioned the uncertainty regarding ice stream shear margin behaviour in time and space and will wait for observational data from the shear margins to be analysed.

I do hope that all these rambles are useful
Dave
A: Thank you again, we appreciate the effort and look forward to advances in laboratory experiments hopefully reproducing the observational data.

References:
Kuiper, E.-J. N., Weikusat, I., de Bresser, J. H. P., Jansen, D., Pennock, G. M., and Drury, M. R.: Using a composite flow law to model deformation in the NEEM deep ice core, Greenland – Part 1: The role of grain size and grain size distribution on deformation of the upper 2207 m, The Cryosphere, 14, 2429–2448, https://doi.org/10.5194/tc-14-2429-2020, 2020a.

Kuiper, E.-J. N., de Bresser, J. H. P., Drury, M. R., Eichler, J., Pennock, G. M., and Weikusat, I.: Using a composite flow law to model deformation in the NEEM deep ice core, Greenland – Part 2: The role of grain size and premelting on ice deformation at high homologous temperature, The Cryosphere, 14, 2449–2467, https://doi.org/10.5194/tc-14-2449-2020, 2020b.

Llorens, M.-G., Griera, A., Bons, P. D., Weikusat, I., Prior, D. J., Gomez-Rivas, E., de Riese, T., Jimenez-Munt, I., García-Castellanos, D., and Lebensohn, R. A.: Can changes in deformation regimes be inferred from crystallographic preferred orientations in polar ice?, The Cryosphere, 16, 2009–2024, https://doi.org/10.5194/tc-16-2009-2022, 2022.

Westhoff, Julien, Nicolas Stoll, Steven Franke, Ilka Weikusat, Paul Bons, Johanna Kerch, Daniela Jansen, Sepp Kipfstuhl, and Dorthe Dahl-Jensen. "A stratigraphy-based method for reconstructing ice core orientation." *Annals of Glaciology* 62, no. 85-86 (2021): 191-202.

Bestmann, M., Piazolo, S., Spiers, C. J., and Prior, D. J., 2005, Microstructural evolution during initial stages of static recovery and recrystallization: new insights from in-situ heating experiments combined with electron backscaVer diffraction analysis: Journal Of Structural Geology, v. 27, no. 3, p. 447-457.

Bestmann, M., and Prior, D. J., 2003, Intragranular dynamic recrystallization in naturally deformed calcite marble: diffusion accommodated grain boundary sliding as a result of subgrain rotation recrystallization: Journal of Structural Geology, v. 25, no. 10, p. 15ti7-1613.

Bouchez, J. L., and Duval, P., 1982, The fabric of polycrystalline ice deformed in simple shear - experiments in torsion, natural deformation and geometrical interpretation: Textures and Microstructures, v. 5, no. 3, p. 171-190.Chauve, T., Montagnat, M., Piazolo, S., Journaux, B., Wheeler, J., Barou, F., Mainprice, D., and Tommasi, A., 2017, Non-basal dislocations should be accounted for in simulating ice mass flow: Earth and Planetary Science LeVers, v. 473, p. 247-255.

Craw, L., 2016, Constrictive Deformation in Ice: A Glaciological Approach [Honours Honours Thesis]: University of Otago, 6ti p.

Craw, L., Qi, C., Prior, D. J., Goldsby, D. L., and Kim, D., 2018, Mechanics and microstructure of deformed natural anisotropic ice: Journal of Structural Geology, v. 115, p. 152-166.

Cross, A. J., Hirth, G., and Prior, D. J., 2017, Effects of secondary phases on crystallographic preferred orientations in mylonites: Geology, v. 45, no. 10, p. ti55-ti58.

Fan, S., Hager, T. F., Prior, D. J., Cross, A. J., Goldsby, D. L., Qi, C., Negrini, M., and Wheeler, J., 2020, Temperature and strain controls on ice deformation mechanisms: insights from the microstructures of samples deformed to progressively higher strains at −10, −20 and −30°C: The Cryosphere, v. 14, no. 11, p. 3875-3ti05.

Fan, S., Wheeler, J., Prior, D. J., Negrini, M., Cross, A. J., Hager, T. F., Goldsby, D. L., and Wallis, D., 2022, Using Misorientation and Weighted Burgers Vector Statistics to Understand Intragranular Boundary Development and Grain Boundary Formation at High Temperatures: Journal of Geophysical Research-Solid Earth, v. 127, no. 8.

Guillope, M., and Poirier, J. P., 197ti, Dynamic Recrystallization During Creep of Single-Crystalline Halite - Experimental-Study: Journal of Geophysical Research, v. 84, no. NB10, p. 5557-5567.

Halfpenny, A., Prior, D. J., and Wheeler, J., 2006, Analysis of dynamic recrystallization and nucleation in a quartzite mylonite: Tectonophysics, v. 427, no. 1-4, p. 3-14.

Jacka, T. H., and Maccagnan, M., 1984, Ice crystallographic and strain rate changes with strain in compression and extension: Cold Regions Science and Technology, v. 8, no. 3, p. 26ti-286.

Jessell, M. W., 1986, Grain-Boundary Migration and Fabric Development in Experimentally Deformed Octachloropropane: Journal of Structural Geology, v. 8, no. 5, p. 527-542.

Journaux, B., Chauve, T., Montagnat, M., Tommasi, A., Barou, F., Mainprice, D., and Gest, L., 2019, Recrystallization processes, microstructure and crystallographic preferred orientation evolution in polycrystalline ice during high-temperature simple shear: The Cryosphere, v. 13, no. 5, p. 14ti5-1511.

Law, R. D., 19ti0, Crystallographic fabrics: a selective review of their applications to research in structural geology, in Knipe, R. J., and RuVer, E. H., eds., Deformation Mechanisms, Rheology and Tectonics, Volume 54: London, Geological Society of London, p. 335-352.

Montagnat, M., Chauve, T., Barou, F., Tommasi, A., Beausir, B., and Frassengeas, C., 2015, Analysis of dynamic recrystallisation of ice from EBSD orientation mapping: Frontiers of Earth Science, v. 3, p. 13.

Poirier, J. P., and Guillope, M., 197ti, Deformation Induced Recrystallization of Minerals: Bulletin de Mineralogie, v. 102, no. 2-3, p. 67-74.

Poirier, J. P., and Nicolas, A., 1975, Deformation-Induced Recrystallization Due to Progressive Misorientation of Subgrains, with Special Reference to Mantle Peridotites: Journal of Geology, v. 83, no. 6, p. 707-720.

Qi, C., Goldsby, D. L., and Prior, D. J., 2017, The down-stress transition from cluster to cone fabrics in experimentally deformed ice: Earth and Planetary Science LeVers, v. 471, p. 136-147.Qi, C., Prior, D. J., Craw, L., Fan, S., Llorens, M. G., Griera, A., Negrini, M., Bons, P. D., and

Goldsby, D. L., 2019, Crystallographic preferred orientations of ice deformed in direct-shear experiments at low temperatures: The Cryosphere, v. 13, no. 1, p. 351-371.

Ree, J. H., 19ti4, Grain-Boundary Sliding and Development of Grain-Boundary Openings in Experimentally Deformed Octachloropropane: Journal of Structural Geology, v. 16, no. 3, p. 403-418.

Schmid, S. M., and Casey, M., 1986, Complete fabric analysis of some commonly observed quartz c-axis patterns, in Hobbs, B. E., and Heard, H. C., eds., Mineral and Rock Deformation: Laboratory Studies - The Paterson Volume, Volume 36, American Geophysical Union, p. 246-261.

Stoll, N., Eichler, J., Hörhold, M., Erhardt, T., Jensen, C., and Weikusat, I., 2021, Microstructure, micro-inclusions, and mineralogy along the EGRIP ice core - Part 1: Localisation of inclusions and deformation patterns: Cryosphere, v. 15, no. 12, p. 5717-5737.

Toy, V. G., Prior, D. J., and Norris, R. J., 2008, Quartz fabrics in the Alpine Fault mylonites: Influence of pre-existing preferred orientations on fabric development during progressive uplift: Journal Of Structural Geology, v. 30, no. 5, p. 602-621.

Trimby, P. W., Prior, D. J., and Wheeler, J., 19ti8, Grain boundary hierarchy development in a quartz mylonite: Journal of Structural Geology, v. 20, no. 7, p. ti17-ti35.

Urai, J. L., Means, W. D., and Lister, G. S., 1986, Dynamic recrystallization of Minerals, in Hobbs, B. E., and Heard, H. C., eds., Mineral and Rock Deformation (Laboratory Studies), Volume 36, p. 161-200.

Wang, Q., Fan, S., Richards, D. H., Worthington, R., Prior, D. J., and Qi, C., 2024, Evolution of crystallographic preferred orientations of ice sheared to high strains by equal-channel angular pressing: EGUsphere, v. 2024, p. 1-34.

Weikusat, I., Jansen, D., Binder, T., Eichler, J., Faria, S. H., Wilhelms, F., Kipfstuhl, S., Sheldon, S., Miller, H., Dahl-Jensen, D., and Kleiner, T., 2017a, Physical analysis of an Antarctic ice core-towards an integration of micro- and macrodynamics of polar ice: Philosophical Transactions of the Royal Society a-Mathematical Physical and Engineering Sciences, v. 375, no. 2086.

Weikusat, I., Kuiper, E. J. N., Pennock, G. M., Kipfstuhl, S., and Drury, M. R., 2017b, EBSD analysis of subgrain boundaries and dislocation slip systems in Antarctic and Greenland ice: Solid Earth, v. 8, no. 5, p. 883-8ti8.

シュウジ, フ., 1987, Orientation of the 700-m mizuho core and its strain history: Department of Applied Physics, Faculty of Engineering, Hokkaido University.

---

## Author Response (AR2)

**Authors response**
We are thankful for the suggestions and implemented them all.

*line numbers with respect to track-changes version of manuscript

L7: add 'the' and 'core' here: '…throughout the 2663-m core, …'; consider removing 'setting' for clarity; add comma after 'stream' (or 'setting' if you choose to keep it)

L16-17: The second half of this sentence is too vague to understand its meaning. What about the 'even older ice from the Eemian' agrees with the microstructural data?

L80: the choice of 'erect' as the verb here sounds odd; consider using something like 'construct' or 'build' here instead

L93: It's uncommon to use 'lower' as an adjective for 'width'. Consider using 'smaller' or 'narrower' for the upstream width here.

Figure 1 caption: Should be 'derived from' here; add comma after May 17

L136: either remove 'so far' or change 'were' to 'have been' to keep the same tense

L146: I don't think 'e.g.' is correct here because you are giving a further example of a noun just before 'e.g.' Either remove it, or add something like 'physical properties' just before it.

L174: It's not clear what you're referring to when you state 'We here extend this to DEP data…'. Extend what?

L188: Use commas instead of parentheses between the authors and publication date for the references within parentheses here.

L194: It would be helpful to restate which records are 'aligned' here to remind the reader after the previous sentences describing the methodology references.

L198-202 (and Figure 2): It would be helpful to define the GI acronym/naming scheme somewhere in here.

L219: 'strengthens' should be singular here

L257: remove extra ')' here

L229: 'Despite a high sampling resolution of measurements up to every metre in this depth regime…' would be a clearer way to state this

L234-235: Wouldn't smaller grain sizes give you more grains and their c-axes to measure?

L239: reword to something like: 'At the depth interval of 2608-2618 m, crystals are several centemetres in length (see section 3.4), and therefore do not have a sufficient number of crystals to measure in the samples. We combined data from these samples with that from six adjacent samples to yield suitable statistics with several hundreds of data points.'

L243: Stay consistent with units here. In the Methods section, you describe the samples at 92 x 70 mm

L244: Within the text, refer to Figures without abbreviation, and abbreviated when in parentheses as a reference.

L251: move ', respectively,' to after the reported eigenvalue values here.

Figure 7 caption: I think you mean 'extend' instead of 'prolong' here.

L343: I think you mean 'for example, non-basal plane deformation is well known for quartz...' here.

L344: add 'is' before 'rarely'

L391: capitalize 'glacial' here

L403: change the semicolon to a comma

L447: 'comparable' should be 'comparably'

L493: either 'In both cores,...' or 'At both core sites,...'

L496: capitalize 'glacial' here to be consistent

L521: change the semicolon here to 'and'

L589: I think you mean 'extend' instead of 'prolong' here too.